# THE SEISMIC WAVEFIELD COMMON TASK FRAMEWORK

**Alexey Yermakov**[1,2], **Yue Zhao**[3], **Marine Denolle**[4], **Yiyu Ni**[4],

**Philippe M. Wyder**[2,5], **Judah Goldfeder**[6], **Stefano Riva**[7], **Jan Williams**[8],

**David Zoro**[1], **Amy Sara Rude**[2], **Matteo Tomasetto**[9], **Joe Germany**[10],

**Joseph Bakarji**[11], **Georg Maierhofer**[12], **Miles Cranmer**[12], **J. Nathan Kutz**[1,2] *

[1]Department of Electrical and Computer Engineering, University of Washington, Seattle, WA 98195
[2]Department of Applied Mathematics, University of Washington, Seattle, WA 98195
[3]High Performance Machine Learning team, SURF, Amsterdam, the Netherlands
[4]Department of Earth and Space Sciences, University of Washington, Seattle, WA 98195
[5]Distyl AI, New York, NY 10016
[6]Department of Computer Science, Columbia University, New York, NY 10027
[7]Department of Energy, Nuclear Engineering Division, Politecnico di Milano, Milan, Italy
[8]Department of Mechanical Engineering, University of Washington, Seattle, WA 98195
[9]Department of Mechanical Engineering, Politecnico di Milano, Milan, Italy
[10]Department of Mathematics, American University in Beirut, Beirut, Lebanon
[11]Department of Mechanical Engineering, American University in Beirut, Beirut, Lebanon
[12]Department of Applied Mathematics and Theoretical Physics, University of Cambridge, Cambridge, UK

## ABSTRACT

Seismology faces fundamental challenges in state forecasting and reconstruction (e.g., earthquake early warning and ground motion prediction) and managing the parametric variability of source locations, mechanisms, and Earth models (e.g., subsurface structure and topography effects). Addressing these with simulations is hindered by their massive scale, both in synthetic data volumes and numerical complexity, while real-data efforts are constrained by models that inadequately reflect the Earth's complexity and by sparse sensor measurements from the field. Recent machine learning (ML) efforts offer promise, but progress is obscured by a lack of proper characterization, fair reporting, and rigorous comparisons. To address this, we introduce a Common Task Framework (CTF) for ML for seismic wavefields, demonstrated here on three distinct wavefield datasets. Our CTF features a curated set of datasets at various scales (global, crustal, and local) and task-specific metrics spanning forecasting, reconstruction, and generalization under realistic constraints such as noise and limited data. Inspired by CTFs in fields like natural language processing, this framework provides a structured and rigorous foundation for head-to-head algorithm evaluation. We evaluate various methods for reconstructing seismic wavefields from sparse sensor measurements, with results illustrating the CTF's utility in revealing strengths, limitations, and suitability for specific problem classes. Our vision is to replace ad hoc comparisons with standardized evaluations on hidden test sets, raising the bar for rigor and reproducibility in scientific ML.

---

*Corresponding author: `kutz@uw.edu`

# 1 INTRODUCTION

Earthquake-induced hazards, tragically illustrated by events like the 1971 M6.7 San Fernando, the 1999 M5.9 Athens (143 fatalities), and the 2011 M5.8 Virginia (up to $300M in damage) earthquakes, are among the most challenging domains for prediction due to the complexity of the dynamics of shaking and ground failures. The physics of seismic wavefields is governed by the 3D elastodynamic wave equation, which must be solved in media whose elastic properties vary sharply in space, depth, and sometimes time. Even small-scale heterogeneities distort and scatter waves—creating highly nonstationary, multi-frequency, and multi-path signals that are far more complex than those produced by uniform or low-dimensional dynamical systems. These wavefields matter at societal scales, where small variations in near-surface structure strongly affect shaking intensity and earthquake hazard, and at planetary scales, where subtle features of global wave propagation reveal the structure and dynamics of Earth's deep interior.

Computational simulations of seismic wavefields must accommodate high-dimensional heterogeneous media, with the numerical complexity increasing with the frequencies that need to be resolved for effective hazard mitigation. At the same time, the rise of distributed acoustic sensing (DAS) and related fiber-optic technologies has made it possible to record dense, unaliased seismic wavefields over kilometer-scale arrays, exposing unprecedented detail in wave propagation, scattering, and mode conversions. These observational advances amplify both the opportunity and the challenge: they provide richly sampled data that can constrain models more tightly than ever before, but they also highlight the limitations of current forward simulations and inversion pipelines. Consequently, the seismological community has begun to investigate machine learning (ML) and artificial intelligence (AI) techniques to accelerate the accurate reconstruction of wavefields from simulations and data. Early results indicate that AI-accelerated techniques can advance the probabilistic modeling of earthquake ground motions. Recent AI methods for wavefield modeling include Neural Operators (e.g., Yang et al., 2023; Zou et al., 2024; Huang & Alkhalifah, 2025; Kong et al., 2025; Lehmann et al., 2024b), Physics-Informed Neural Networks (Moseley et al., 2023), combinations of both (Huang et al., 2025), established general deep learning architectures (Lyu et al., 2025; Nakata et al., 2025), and reduced-order models that leverage many simulations to discover Proper Orthogonal Decomposition and function bases for low-cost wavefield reconstruction (Rekoske et al., 2023; 2025).

The rapid development and adoption of these methods on seismological data and simulations have outpaced efforts to compare them objectively. In the absence of a common evaluation standard, new methods are not assessed fairly against existing approaches, resulting in weak baselines, reporting bias, and inconsistent evaluations (McGreivy & Hakim, 2024; Wyder et al.). Few scientific and engineering domains have mitigated these problems, instead relying on self-reporting by providing both training and testing datasets to the community. While reducing the evaluation burden on the original authors, self-reporting opens the door to problematic practices such as *p*-hacking and implicit optimization on the test set. Only with a truly withheld test set is a rigorous and impartial comparison among methods possible.

## 1.1 RELATED WORKS

Common task frameworks (CTFs) have been pivotal in driving and quantifying progress in ML and AI (Donoho, 2017). Landmark CTFs catalyzed key advances: ImageNet (Deng et al., 2009) provided the stage that revitalized Convolutional Neural Networks, with Krizhevsky et al. (2012) demonstrating their superiority over classical methods in large-scale image recognition. Natural-language challenges, ranging from code generation (Chen et al., 2021) to formal reasoning and mathematics (Hendrycks et al., 2021; Cobbe et al., 2021), have been central to advancing large language models, with recent systems such as DeepSeek-R1-Zero demonstrating competitive performance alongside state-of-the-art models (DeepSeek-AI et al., 2025). Competitive games have served as another CTF: matches in Go and shogi provided the testbed that led to AlphaZero (Silver et al., 2018). Video game environments such as the Arcade Learning Environment (Bellemare et al., 2013), based on Atari 2600 games, enabled the breakthrough results of deep Q-networks (Mnih et al., 2015). Finally, platforms providing models with control inputs, such as the OpenAI Gym (Brockman et al., 2016) and MuJoCo (Todorov et al., 2012), have accelerated the development and testing of reinforcement-learning methods at scale. Mature CTF platforms are, therefore, critical driving forces of innovation and progress.

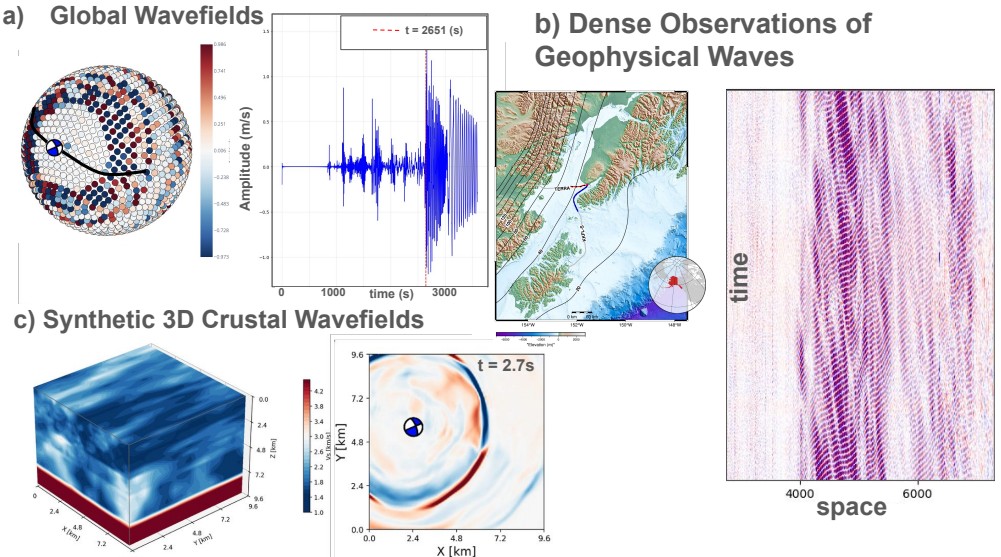

Figure 1: The Seismic Wavefield CTF scores the performance of methods on (a) global wavefields from sparse sensor measurements, (b) dense observations of real geophysical wavefields from Distributed Acoustic Sensing, and (c) dense simulations of 3D crustal wavefields. These three datasets represent a broad range of challenging datasets encountered frequently by seismologists and present challenges for models in forecasting and state reconstruction.

Despite these successes, many domains beyond the major areas of computer vision, natural language processing, and reinforcement learning still suffer from a lack of a community standard for fairly evaluating new methods. Indeed, aside from *critical assessment of protein structure prediction* CASP (predictioncenter.org) (Donoho, 2017), science and engineering have largely ignored CTFs and have instead relied on self-reporting benchmarks. Recent work from Wyder et al. has begun to bridge this gap by providing a CTF for scientific ML models on canonical nonlinear systems. But substantial work remains: discipline-specific CTF suites, standardized evaluation protocols (including withheld test sets), and community-maintained leaderboards are needed to ensure fair comparison, reproducibility, and sustained progress.

## 1.2 THE SEISMIC WAVEFIELD COMMON TASK FRAMEWORK

We propose a CTF for seismology that is, in its initial release, primarily focused on evaluating ML and AI algorithms modeling the seismic wavefields shown in Fig. 1. Based on the work from Wyder et al., this CTF provides training datasets with explicit tasks related to forecasting and reconstruction under various challenges, such as noisy measurements, limited data, and varying system parameters. The datasets are limited in volume by design, in order to democratize access to the exercise. Participants submit predictions for a withheld test set over specified timesteps. The predictions are evaluated and scored on a diverse set of metrics by an independent referee and posted on a leaderboard.

Scoring is by nature reductive—reducing a method's performance to a single floating point value. We adopt a multi-metric scoring scheme because a single score is often insufficient to characterize suitability for different scientific uses. As a result, we use the carefully designed twelve-score system from Wyder et al. which maps to critical tasks for seismic wavefield data objectively. A summary, or composite score, is also computed that gives the overall score for a given method. Task-specific and overall rankings are highlighted in this paper and displayed on a leaderboard.

For each submission, we calculate the full suite of twelve task scores. These results quickly convey model strengths and weaknesses—e.g., robustness to noise, performance in limited-data regimes, or parametric generalization—so users can choose methods suited to their needs. The composite score is the mean of the task scores; using multiple task-specific metrics avoids a winner-take-all outcome and promotes methods that are fit-for-purpose across diverse seismological applications.

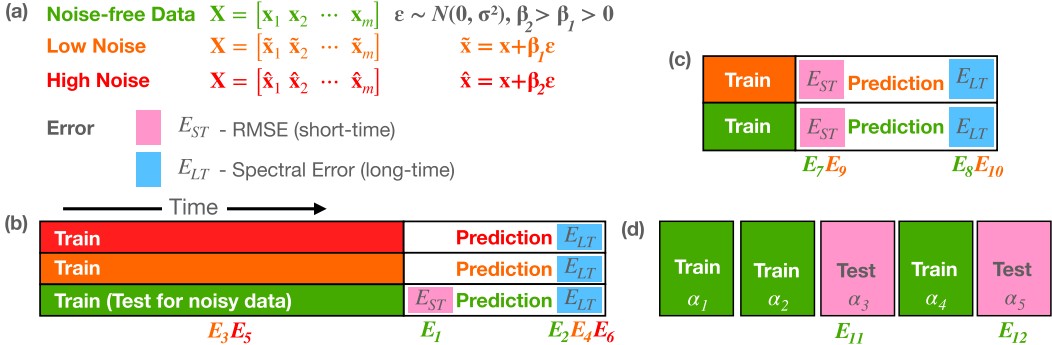

Figure 2: The Seismic Wavefield CTF scores the performance of methods on seismic wavefield datasets. (a) Data is collected and organized into matrices, which are then split into testing and training sets. RMSE errors are computed for reconstruction and short-time forecasting, while the spectral error computes the statistics of long-time forecasting (spatial or temporal). (b) Forecasting and reconstruction tasks are evaluated on noise-free, low-noise, and high-noise data. Methods are also evaluated when (c) only limited data is available and (d) for reconstruction of parametrically dependent data.

Once the Seismic Wavefield CTF is launched on Kaggle[1], we invite the community to evaluate their methods on the CTF by taking the following steps:

1. Sign up and sign in on Kaggle
2. Train your model with our training data and generate predictions for each task
3. Submit prediction files to the competition platform
4. See your score on the leaderboard

To interact with the Seismic Wavefield CTF before the competition launches, visit our GitHub repository[2], install the Python package, and evaluate your method on our publicly available datasets. Our dataset and Python package don't require high-performance hardware and can be run on a laptop.

## 2 DATASETS & EVALUATION METRICS

We extend the CTF with three new challenging seismic wavefield datasets and evaluate two of them, the global wavefields dataset and the DAS dataset, on several commonly used models in scientific machine learning. A visualization of the spatio-temporal structure of these wavefields is in Fig. 1. All datasets exhibit complex and challenging behavior for the tasks of reconstruction and forecasting under the constraints of noise, limited data, and parametric dependence. While these datasets serve as a starting point, the CTF will evolve to include both more complex data and more challenging tasks. The Seismic Wavefield CTF is a sustainable platform that evolves and grows as the community develops more sophisticated methods and algorithms and faces new challenges.

### 2.1 SEISMIC WAVEFIELDS AS SPATIO-TEMPORAL SYSTEMS

Seismic wavefields are the response to the elastodynamic wave equation in Earth models of elastic properties that vary in space. When the Earth models are uniform, solutions are simple spatiotemporal fields of ballistic waves: P, S, and, when the Earth's surface is included as a traction-free boundary, surface Rayleigh and Love waves. Spatial and depth variations in Earth properties distort and scatter the wavefields, yielding great complexity in the spatio-temporal pattern of seismic wavefields, with numerical complexity that scales with resolvable seismic frequencies and domain size, the two main bottlenecks in predicting ground motions relevant for seismic hazard analysis. The three data sets we provide are different in scale and scope: the first two are products of numerical simulations at

---

[1]We are proposing a launch date of March 1, 2026, on Kaggle
[2]Available at `https://github.com/CTF-for-Science/ctf4science`

the global and crustal scales and target different goals and research audience (an earthquake hazard specialist or a geodynamicist). The third data set is purely observational from a sensor array and will target an audience of oceanographers and marine geophysicists.

The first dataset, for which we present results in the leaderboard, is a dataset of globally propagating seismic waves (Fig. 1-a). van Driel et al. (2015) developed an efficient workflow to generate and store Green's functions that can be reused for arbitrary source locations and receivers on the Earth's surface using the AxiSEM numerical solver (Nissen-Meyer et al., 2014). Such a framework has enabled the vast dissemination of complex global wavefields, for Earth and Mars structures, democratizing access for the scientific community to modeled full wavefields (Krischer et al., 2017), which would be otherwise computationally intractable for most scientists. We leverage this database of Green's functions to construct a series of earthquake wavefields recorded at 2048 sensors located at the surface of a sphere. The wavefields are computed in the IASP91 Earth model that was designed to match the arrival time of P and S waves (Kennett & Engdahl, 1991) and has become a standard of radially symmetric Earth models, where seismic properties (P wavespeed, S wavespeed, and Earth material density) increase as a function of depth. Our dataset is built upon these Green's functions, which, when convolved with a source mechanism, deliver a realistic seismogram. We choose 2048 sensors distributed on a Fibonacci sphere of 6371 km radius, a sampling rate of 1 Hz (resampled from the original simulations that resolved as short as 2-second period waves), and time series up to 3600 s. Each dataset file contains NumPy arrays with shape (time_steps, 2048) representing the vertical z-component of velocity seismograms. Given that realistic waveform data span many orders of magnitude, we normalize the dataset to have zero mean and unit variance per earthquake event, ensuring temporal continuity and predictability for each task.

The second dataset uses a novel geophysical sensing technology that leverages optical scattering to transform telecommunication fibers into arrays of virtual sensors. Referred to as Distributed Acoustic Sensing (DAS) (Fig. 1-b), this technology is revolutionizing the observations of earthquakes (e.g., Yin et al., 2023), marine mammals (e.g., Wilcock et al., 2023), ocean dynamics (Lindsey et al., 2019), and structural health of offshore wind turbines. Major challenges for DAS are the massive data volumes (Spica et al., 2023; Ni et al., 2023) and the complexity of the wavefields (Xu et al., 2025), motivating AI-based compression and reconstruction (e.g., Ni et al., 2024). We prepared a dataset of ten 1-minute recordings sampled at 5 Hz and low-pass filtered up to 1 Hz. The data present an interesting overlap between earthquake and oceanographic signals at multiple spatial and temporal scales. The channel spacing is 9.57 m, and we trimmed the data files to 3000 channels among the 9000 channels available on that particular fiber. Typical waves that dominate the shallow offshore DAS are the surface swell that follows the dispersion relation $\omega^2 = gk \tanh kh$, where $\omega$ is the angular frequency, $g$ is the gravity constant, $k$ is the wavenumber, and $h$ is the water depth. For the test set provided, $h \sim 30m$, making surface gravity waves particularly dispersive in the data. This dataset is purely observational.

The third dataset (Fig. 1-c) is an extension of the curated datasets from Lehmann et al. (2024a). It comprises synthetic 3D seismic wavefields in a heterogeneous 3D crustal model. Earthquakes were modeled as point sources with a double-couple mechanism represented by six parameters; source location and focal mechanism (strike, dip, and rake angle) were drawn at random within the model volume. The modeled spatial and temporal scales are aligned with those used in recent AI models of wavefields (e.g., Kong et al., 2025; Rekoske et al., 2025), relevant for seismic analysis of crustal earthquakes that can pose substantial risk to people and infrastructure when they occur in populated areas. Ten independent simulations were produced. Each simulation yields three-component velocity seismograms on a $32 \times 32 \times 32$ heterogeneous grid. Virtual sensors form a $94 \times 94$ grid arranged on the top of the model volume with 100 m spacing. These seismograms are sampled for 6 seconds at 50 Hz.

## 2.2 METRICS

### 2.2.1 FORECASTING (2 SCORES)

The first set of tasks, shown in Fig. 2-b, involves the approximation of the future state of the system. Thus, given a data matrix representing the dynamics over $t \in [0, 4T]$ ($\mathbf{X}_1 \in \mathbb{R}^{4m \times n}$), a generated forecast is requested from the model being tested for $t \in [4T, 6T]$ ($\mathbf{X}_{1pred} \in \mathbb{R}^{2m \times n}$) to compare with the ground-truth ($\mathbf{X}_{1test} \in \mathbb{R}^{2m \times n}$), with $n$ being the dimension of the system and $m$ being the

number of forecasted time steps. The forecasting score is composed of two scores evaluating both the short-time forecast $E_{\text{ST}}$, which is computed using Root Mean Square Error (RMSE) between the test set and the model's approximation, and the long-time forecast $E_{\text{LT}}$, which is computed using the spectral error based upon the power spectral density, see Fig. 2-a. Short-time forecasting measures trajectory accuracy where deterministic prediction is feasible, while long-time forecasting measures statistical fidelity where only the system's broad statistical properties are recoverable.

For the challenge dynamics of interest, the sensitivity to initial conditions is common, making long-range forecasting to match the test set an unreasonable task, given fundamental mathematical limitations with Lyapunov times. Thus, the long-time error is computed by least-squares fitting of the power spectrum $\mathbf{P}(\mathbf{X}, \mathbf{k}, k) = \ln(|\text{FFT}(\mathbf{X}[-k:-1, \mathbf{k}])|^2)$, where $\texttt{fftshift}$ has been used to model the data in the wavenumber domain and $\mathbf{k} = n/2 - k_{max} : n/2 + (k_{max} + 1)$ with $k_{max} = 100$. This means that we look at the match in the first 100 wavenumbers of the power spectrum over a long-time simulation. Let $\hat{\mathbf{X}}$ be the ground-truth matrix, $\tilde{\mathbf{X}}$ be the prediction matrix, and $k \in (0, T)$ an integer specifying how to split the matrices for the short-time and long-time scores. The following two error scores are then computed:

$$S_{\text{ST}}(\tilde{\mathbf{X}}, \hat{\mathbf{X}}) = \frac{\|\hat{\mathbf{X}}[1:k,:] - \tilde{\mathbf{X}}[1:k,:]\|}{\|\hat{\mathbf{X}}[1:k,:]\|}, \tag{1}$$

$$S_{\text{LT}}(\tilde{\mathbf{X}}, \hat{\mathbf{X}}) = \frac{\|\mathbf{P}(\hat{\mathbf{X}}, \mathbf{k}, k) - \mathbf{P}(\tilde{\mathbf{X}}, \mathbf{k}, k)\|}{\|\mathbf{P}(\hat{\mathbf{X}}, \mathbf{k}, k)\|}. \tag{2}$$

It is clear that there are many ways to evaluate the long-range forecasting capabilities. We followed in the footsteps of Wyder et al. and chose a simple and transparent metric, fully understanding that more nuanced scoring could be used. To provide a reasonable range, we then compute the two scores

$$E_1 = 100(1 - S_{\text{ST}}(\mathbf{X}_{1pred}, \mathbf{X}_{1test})), \quad E_2 = 100(1 - S_{\text{LT}}(\mathbf{X}_{1pred}, \mathbf{X}_{1test})), \tag{3}$$

meaning in each case a score of $E_i = 100$ corresponds to a perfect match. Note that, as a baseline, a solution guess of zeros $\tilde{\mathbf{X}}[1:k,:] = \mathbf{0}$ (corresponding also to $\mathbf{P}(\tilde{\mathbf{X}}, \mathbf{k}, k) = \mathbf{0}$) gives a score of $E_1 = E_2 = 0$.

**Input:** $\mathbf{X}_{1train} \in \mathbb{R}^{4m \times n}$; **Output:** $\mathbf{X}_{1pred} \in \mathbb{R}^{2m \times n}$; **Scores:** $E_1, E_2$.

### 2.2.2 NOISY DATA (4 SCORES)

The ability to handle noise is critical in all data-driven applications as sensors and measurement technologies are by default embedded with varying levels of noise. Methods that work with numerically accurate data, for example, data points that are $10^{-6}$ accurate, may be useful for model reduction, but are rarely suitable for discovery and engineering design from real-world data. Both strong and weak noise are considered, as these represent realistic challenges to be addressed in practice.

This task is very similar to the forecasting described above, but now with noise added to the data. Specifically, the model is provided data matrices $\mathbf{X}_{2train} \in \mathbb{R}^{4m \times n}$ and $\mathbf{X}_{3train} \in \mathbb{R}^{4m \times n}$ representing the evolution with low or high noise respectively. The objective is to first produce a reconstruction of the data itself, i.e. denoise the data to produce an estimate of the true state of the dynamics, $\mathbf{X}_{2pred}, \mathbf{X}_{4pred} \in \mathbb{R}^{4m \times n}$ for $\mathbf{X}_{2train}, \mathbf{X}_{3train}$ respectively, and the second objective is to then forecast the future state, matrices $\mathbf{X}_{3pred}, \mathbf{X}_{5pred} \in \mathbb{R}^{2m \times n}$ for $\mathbf{X}_{2train}, \mathbf{X}_{3train}$ respectively. For the reconstruction objective, a least-square fit is used between the approximation of the denoised data and the truth, and for the forecasting objective, a long-time evaluation is computed, leading to the following scores:

$$E_3 = 100(1 - S_{\text{ST}}(\mathbf{X}_{2pred}, \mathbf{X}_{2test})), \quad E_4 = 100(1 - S_{\text{LT}}(\mathbf{X}_{3pred}, \mathbf{X}_{3test})),$$
$$E_5 = 100(1 - S_{\text{ST}}(\mathbf{X}_{4pred}, \mathbf{X}_{4test})), \quad E_6 = 100(1 - S_{\text{LT}}(\mathbf{X}_{5pred}, \mathbf{X}_{5test})).$$

**Input:** $\mathbf{X}_{2train}, \mathbf{X}_{3train} \in \mathbb{R}^{4m \times n}$; **Output:** $\mathbf{X}_{2pred}, \mathbf{X}_{4pred} \in \mathbb{R}^{4m \times n}$, $\mathbf{X}_{3pred}, \mathbf{X}_{5pred} \in \mathbb{R}^{2m \times n}$; **Scores:** $E_3, E_4, E_5, E_6$.

### 2.2.3 LIMITED DATA (4 SCORES)

Data limitations are common in real-world physical systems and often affect the success of data-driven methods. Thus, testing for model performance on low data is critically important and provides important insight for potential users.

Figure 2-c demonstrates the nature of the task. In this case, only a limited number of snapshots $M$ of numerically accurate data are given $\mathbf{X}_{4train} \in \mathbb{R}^{M \times n}$. From this limited data, a forecast must be made which is evaluated with both error metric equation 1 and equation 2 on the approximated future $\mathbf{X}_{6pred} \in \mathbb{R}^{2m \times n}$. The experiment is repeated with noise on the measurements using the training matrix $\mathbf{X}_{5train} \in \mathbb{R}^{M \times n}$ for which a forecasting prediction matrix is produced $\mathbf{X}_{7pred} \in \mathbb{R}^{2m \times n}$. The performance is evaluated on the following scores representing short and long-time metrics for both noise-free and noisy data, respectively.

$$E_7 = 100(1 - S_{\text{ST}}(\mathbf{X}_{6pred}, \mathbf{X}_{6test})), \quad E_8 = 100(1 - S_{\text{LT}}(\mathbf{X}_{6pred}, \mathbf{X}_{6test})),$$
$$E_9 = 100(1 - S_{\text{ST}}(\mathbf{X}_{7pred}, \mathbf{X}_{7test})), \quad E_{10} = 100(1 - S_{\text{LT}}(\mathbf{X}_{7pred}, \mathbf{X}_{7test})).$$

Two error scores (analogous to $E_1$ and $E_2$) are produced for the noise-free and noisy limited data. These scores are $E_7$ (short-time forecast) and $E_8$ (long-time forecast) for the noise free case and $E_9$ (short-time forecast) and $E_{10}$ (long-time forecast) for the noisy case.

**Input:** $\mathbf{X}_{4train}, \mathbf{X}_{5train} \in \mathbb{R}^{M \times n}$; **Output:** $\mathbf{X}_{6pred}, \mathbf{X}_{7pred} \in \mathbb{R}^{2m \times n}$;
**Scores:** $E_7, E_8, E_9, E_{10}$.

### 2.2.4 PARAMETRIC GENERALIZATION (2 SCORES)

Finally, the ability of a model to generalize to different parameter values is evaluated. For this case, the model's ability to interpolate and extrapolate to new parameter regimes is considered with noise-free data. The interpolation and extrapolation are each their own score, resulting in two scores that evaluate parametric dependence.

Figure 2-d shows the basic architecture of the task. Three training datasets are provided with three different (unknown) parameter values $\mathbf{X}_{6train}, \mathbf{X}_{7train}, \mathbf{X}_{8train} \in \mathbb{R}^{4m \times n}$. Construction of the dynamics in parametric regimes that are interpolatory $\mathbf{X}_{8pred} \in \mathbb{R}^{2m \times n}$ and extrapolatory $\mathbf{X}_{9pred} \in \mathbb{R}^{2m \times n}$ is required. For both of the tasks, a burn-in matrix of size $M \times n$ (where $M \leq m$) is given ($\mathbf{X}_{9train}$ and $\mathbf{X}_{10train}$ respectively) and the performance is evaluated using the short-time metric equation 1.

$$E_{11} = 100(1 - S_{\text{ST}}(\mathbf{X}_{8pred}, \mathbf{X}_{8test})), \quad E_{12} = 100(1 - S_{\text{ST}}(\mathbf{X}_{9pred}, \mathbf{X}_{9test})).$$

**Input:** $\mathbf{X}_{6train}, \mathbf{X}_{7train}, \mathbf{X}_{8train} \in \mathbb{R}^{4m \times n}, \mathbf{X}_{9train}, \mathbf{X}_{10train} \in \mathbb{R}^{M \times n}$;

**Output:** $\mathbf{X}_{8pred}, \mathbf{X}_{9pred} \in \mathbb{R}^{2m \times n}$; **Scores:** $E_{11}, E_{12}$.

### 2.2.5 COMPOSITE SCORE

We compute a composite score ($AvgScore$) per dataset from metrics $E_1$ through $E_{12}$ by averaging the resulting scores for each method. This score is evaluated per method, not per model. Thus, each method can fit a model for each task and produce the best possible score. All scores are clipped such that $E_i \in [-100, 100]$, thus $AvgScore \in [-100, 100]$. Methods that cannot produce a result for a given task receive the minimum score $-100$.

### 2.3 DATASET DETAILS

In Table 2 we provide a detailed breakdown of the matrices used for training and testing each metric. The specific matrix shapes for the global wavefields dataset, the DAS dataset, and 3D crustal dataset are provided in Table 3. Note that $m = 500$ and $M = 500$ for the global wavefield and DAS dataset, and $m = 250$ and $M = 200$ for the 3D crustal wavefield dataset. In the 3D crustal wavefield dataset, the data is parameterized by the Earth velocity model and the earthquake source (location and mechanism). Such information is not provided in the parametric generalization tasks ($E_{11}$ and $E_{12}$), therefore making matrices $\mathbf{X}_{6,7,8,9,10\text{train}}$ and $\mathbf{X}_{8,9\text{test}}$ smaller compared to matrices in the other tasks.

Table 1: Model Scores on the global wavefields and DAS datasets. The evaluated models are: Chronos (Ansari et al., 2024b), DeepONet (Lu et al., 2021a), FNO (Li et al., 2021), Higher Order DMD (Le Clainche & Vega, 2017a), KAN (Liu et al., 2025b), LLMTime Gruver et al. (2023), LSTM (Hochreiter & Schmidhuber, 1997), Moirai (Liu et al., 2024a), NeuralODE (Ruthotto, 2024), ODE-LSTM (Coelho et al., 2024), Opt DMD (Askham & Kutz, 2018a), Panda Lai et al. (2025), PyKoopman (Brunton et al., 2022; Pan et al., 2024), Reservoir (Jaeger, 2001b; Maass & Markram, 2004; Pathak et al., 2018), SINDy (Brunton et al., 2016; Fasel et al., 2022), SpaceTime (Zhang et al., 2023), Sundial (Liu et al., 2025a), and TabPFN Hoo et al. (2025)

| Global Wavefields Dataset | | | | | | | | | | | | | |
|---|---|---|---|---|---|---|---|---|---|---|---|---|---|
| Model | Avg Score | E1 | E2 | E3 | E4 | E5 | E6 | E7 | E8 | E9 | E10 | E11 | E12 |
| LSTM | 13.18 | -1.11 | 13.8 | 69.7 | **19.84** | 48.83 | 24.32 | -40.07 | 11.34 | -18.38 | **30.97** | **4.09** | -5.22 |
| ODE-LSTM | 5.71 | -0.07 | 6.67 | 65.78 | -20.36 | 41.1 | 24.1 | -67.04 | **13.16** | -0.29 | 5.31 | -0.38 | -5.23 |
| Baseline Average | 0.16 | 0.0 | 0.0 | 0.0 | 0.02 | 3.59 | 0.03 | -1.73 | 0.01 | -0.02 | 0.02 | -0.03 | -0.02 |
| Baseline Zeros | 0.0 | 0.0 | 0.0 | 0.0 | 0.0 | 0.0 | 0.0 | 0.0 | 0.0 | 0.0 | 0.0 | 0.0 | **0.0** |
| Higher Order DMD | 0.0 | 0.0 | 0.0 | -0.01 | 0.0 | -0.02 | 0.0 | 0.0 | 0.02 | 0.0 | 0.0 | 0.0 | **0.0** |
| Reservoir | -14.76 | -0.96 | 6.73 | 75.37 | -100.0 | -100.0 | **33.63** | **6.03** | 1.61 | 1.06 | -100.0 | 1.15 | -1.75 |
| Opt DMD | -25.76 | -2.65 | 9.12 | 2.81 | -91.47 | 0.49 | 2.67 | -36.52 | -100.0 | -10.73 | -82.21 | -0.55 | -0.03 |
| SINDy | -26.44 | 0.0 | 0.02 | -100.0 | 3.89 | -100.0 | 0.13 | -19.12 | 2.69 | -1.65 | -100.0 | -3.12 | -0.08 |
| FNO | -30.92 | **4.91** | -100.0 | **80.82** | -100.0 | **75.81** | -100.0 | -23.72 | -9.36 | -28.82 | -100.0 | -35.29 | -35.4 |
| PyKoopman | -34.48 | -0.16 | 5.08 | 4.51 | -100.0 | -0.21 | -100.0 | 2.2 | 0.21 | -9.95 | -100.0 | -15.44 | -100.0 |
| NeuralODE | -40.89 | -58.12 | **17.81** | -100.0 | -25.08 | -100.0 | 9.49 | -100.0 | 9.95 | -63.55 | 30.73 | -57.47 | -54.39 |
| SpaceTime | -45.19 | -11.2 | -100.0 | 10.5 | -100.0 | -15.75 | -100.0 | -48.9 | 3.26 | -19.18 | -100.0 | -34.2 | -26.83 |
| DeepONet | -50.1 | -100.0 | -100.0 | -100.0 | 0.08 | -100.0 | -100.0 | -6.95 | 5.73 | 0.0 | 0.0 | -0.08 | -100.0 |
| KAN | -63.55 | -0.43 | 3.51 | 3.3 | -100.0 | 1.66 | -7.51 | -100.0 | -100.0 | -100.0 | -100.0 | -100.0 | -100.0 |
| Moirai | -100.0 | -100.0 | -100.0 | -100.0 | -100.0 | -100.0 | -100.0 | -100.0 | -100.0 | -100.0 | -100.0 | -100.0 | -100.0 |
| Panda | -100.0 | -100.0 | -100.0 | -100.0 | -100.0 | -100.0 | -100.0 | -100.0 | -100.0 | -100.0 | -100.0 | -100.0 | -100.0 |
| Chronos | -100.0 | -100.0 | -100.0 | -100.0 | -100.0 | -100.0 | -100.0 | -100.0 | -100.0 | -100.0 | -100.0 | -100.0 | -100.0 |
| TabPFN | -100.0 | -100.0 | -100.0 | -100.0 | -100.0 | -100.0 | -100.0 | -100.0 | -100.0 | -100.0 | -100.0 | -100.0 | -100.0 |
| LLMTime | -100.0 | -100.0 | -100.0 | -100.0 | -100.0 | -100.0 | -100.0 | -100.0 | -100.0 | -100.0 | -100.0 | -100.0 | -100.0 |
| Sundial | -100.0 | -100.0 | -100.0 | -100.0 | -100.0 | -100.0 | -100.0 | -100.0 | -100.0 | -100.0 | -100.0 | -100.0 | -100.0 |

| Distributed Acoustic Sensing Dataset | | | | | | | | | | | | | |
|---|---|---|---|---|---|---|---|---|---|---|---|---|---|
| Model | Avg Score | E1 | E2 | E3 | E4 | E5 | E6 | E7 | E8 | E9 | E10 | E11 | E12 |
| PyKoopman | 12.7 | 7.11 | 14.5 | 33.38 | 0.42 | 71.52 | 3.0 | 7.35 | **25.06** | 2.94 | 2.55 | -15.41 | -0.01 |
| ODE-LSTM | 11.58 | -1.2 | 0.39 | 36.65 | **7.24** | 82.88 | 14.39 | -9.63 | 15.76 | -16.0 | 14.79 | -27.93 | 8.79 |
| Baseline Average | 0.21 | -0.04 | 0.01 | 0.02 | 0.2 | 0.0 | 0.0 | 2.79 | 1.67 | -0.22 | 3.5 | -2.99 | -2.45 |
| Baseline Zeros | 0.0 | 0.0 | 0.0 | 0.0 | 0.0 | 0.0 | 0.0 | 0.0 | 0.0 | 0.0 | 0.0 | 0.0 | 0.0 |
| Sundial | -0.57 | **49.57** | 0.82 | -63.34 | 3.77 | -49.28 | 7.09 | **33.24** | 23.33 | **13.83** | **20.43** | -30.52 | -15.81 |
| TabPFN | -1.68 | -7.09 | -3.32 | **82.28** | -4.75 | 89.83 | 0.16 | -5.94 | -100.0 | -11.66 | -2.03 | -55.8 | -1.83 |
| Higher Order DMD | -3.59 | -0.01 | 0.0 | -0.02 | 0.0 | 0.0 | 0.0 | -1.14 | 0.0 | 0.04 | -0.19 | -41.64 | -0.07 |
| LSTM | -8.0 | -21.7 | 19.39 | 38.47 | -17.39 | 88.96 | 15.87 | -26.4 | -10.61 | -36.66 | -60.52 | -95.21 | 9.81 |
| SINDy | -18.55 | -4.77 | 4.2 | -100.0 | 3.3 | -100.0 | 18.84 | -38.93 | 16.81 | -38.93 | 16.81 | 0.17 | -0.07 |
| FNO | -26.6 | -24.15 | -100.0 | 77.11 | -100.0 | 88.54 | -0.97 | -44.24 | -100.0 | 1.36 | -70.49 | -30.52 | -15.81 |
| KAN | -28.95 | -13.68 | 4.77 | 13.17 | -100.0 | 65.16 | 2.39 | -100.0 | -100.0 | -1.78 | -1.99 | -100.0 | -0.17 |
| NeuralODE | -33.44 | -55.37 | **21.16** | -100.0 | -41.64 | -100.0 | 10.37 | -57.66 | 18.58 | -100.0 | -36.08 | **24.68** | **14.63** |
| Opt DMD | -44.08 | -10.84 | -100.0 | 81.66 | -100.0 | **91.49** | -100.0 | -34.38 | -49.86 | -6.97 | -100.0 | -100.0 | -100.0 |
| DeepONet | -45.11 | -100.0 | -100.0 | 0.05 | -42.97 | 0.03 | -100.0 | -0.05 | -100.0 | 0.03 | 1.61 | -100.0 | 0.0 |
| SpaceTime | -56.04 | -40.51 | -100.0 | -36.97 | -100.0 | 60.73 | -100.0 | -27.47 | -100.0 | -100.0 | -100.0 | -4.35 | -23.91 |
| Moirai | -61.56 | 18.11 | -20.93 | -100.0 | -100.0 | -100.0 | -100.0 | 8.01 | -100.0 | -51.81 | -100.0 | -30.52 | -15.81 |
| Chronos | -87.19 | -100.0 | -100.0 | -100.0 | -100.0 | -100.0 | -100.0 | -100.0 | -100.0 | -100.0 | -100.0 | -30.52 | -15.81 |
| Panda | -98.58 | -100.0 | -100.0 | -100.0 | -100.0 | -100.0 | -100.0 | -100.0 | -100.0 | -100.0 | -100.0 | -82.98 | -100.0 |
| LLMTime | -100.0 | -100.0 | -100.0 | -100.0 | -100.0 | -100.0 | -100.0 | -100.0 | -100.0 | -100.0 | -100.0 | -100.0 | -100.0 |

Our GitHub repository provides the global wavefields, DAS, and 3D crustal wavefields datasets in `.npz` files. The repository contains all the necessary code for data loading, validation split creation, hyperparameter tuning, model inference, and final scoring. For hyperparameter tuning metrics $E_1$ through $E_{10}$, we split the training matrix into two parts: the first 80% is the training data, and the last 20% is the testing data. For hyperparameter tuning metric $E_{11}$ we use matrices $\mathbf{X}_{6,8\text{train}}$ for the training data and $\mathbf{X}_{7\text{train}}$ for the testing data, and the burn-in matrix comes from $\mathbf{X}_{7\text{train}}$ as well. For hyperparameter tuning metric $E_{12}$ we use matrices $\mathbf{X}_{6,7\text{train}}$ for the training data and $\mathbf{X}_{8\text{train}}$ for the testing data, and the burn-in matrix comes from $\mathbf{X}_{8\text{train}}$ as well. More details on hyperparameter tuning can be found in the Appendix subsubsection A.2.1.

Below, we highlight our results on the global wavefields and DAS datasets while reserving the 3D crustal wavefields dataset for the upcoming Kaggle competition mentioned in the introduction.

## 3 METHODS, BASELINES AND RESULTS

We characterize eighteen highly-cited modeling methods on the global wavefields and DAS datasets. Table 1 shows all scored methods and their resulting performance scores. The Seismic Wavefield CTF includes two naive baseline methods: predicting zero and predicting the average. In our evaluations, we use the zero prediction as the reference baseline for both datasets. Due to the dataset having a zero mean after normalization, the average and zero baseline report similar scores.

While some models score high on specific tasks, no model scores high across all tasks. Overall, the results demonstrate that the dataset and specific tasks are challenging enough to produce a distribution of scores that characterizes the methods. A complete overview of all models' performance metrics on the global wavefields dataset can be found in Table 1.

### 3.1 OBSERVATIONS

The results in Table 1 demonstrate that many complex ML architectures fail to outperform the baseline of predicting zeros on the global wavefields dataset. Our multi-score evaluation scheme provides deeper insight into the difficulty of the dataset as well as the strengths and weaknesses of the evaluated models. Notably, many commonly used ML/AI models perform very poorly on the assigned tasks. Although the current CTF provides limited data, the difficulty most methods have in exceeding the zero baseline indicates that considerable research and development are still needed before ML/AI models make a meaningful impact in the field. The overall best models are the RNN-based architectures – the LSTM and ODE-LSTM (Coelho et al., 2024; Hochreiter & Schmidhuber, 1997) – scoring in the top two for both the global wavefields dataset and the DAS dataset. The RNNs perform exceptionally well on tasks $E_3$ and $E_5$, corresponding to the denoising of low- and high-noise data. Their advantage likely stems from a relatively modest parameter count combined with relatively strong expressive power, allowing them to act as implicit regularizers when training on the limited data available for auto-regressive forecasting. In contrast, statistical approaches such as DMD tend to overfit the noise in these datasets, whereas the RNNs, trained with an MSE loss, are more robust. Furthermore, the Sundial foundation model performed best in short-term forecasting for the DAS dataset on metrics E1, E7, and E9. Since none of the models scored above 50 on any of the forecasting metrics, we omit the prediction plots of all models since they show nothing of interest.

This result also demonstrates how the multi-metric scoring in the CTF is better than a single score. While the RNNs achieve the highest average score, they perform exceptionally poorly on tasks $E_7$ and $E_9$, corresponding to short-term prediction on limited data (noiseless and noisy, respectively). This disparity underscores that no single model dominates across all regimes and that task-specific evaluation is essential. We encourage practitioners to consider what properties they value most in a model and to match those with the metrics we provide to identify the model(s) that best serve their purposes. As we've demonstrated, there is no true one-size-fits-all model that performs well across the board.

We present these findings to stimulate discussion within the seismology community about the appropriate choice of ML models for different problem settings and to demonstrate the value of a comprehensive CTF framework. As Table 1 shows, every task on the global seismic wavefields dataset offers room for improvement that can result in real-world benefit. We hope that, as the CTF grows, more models will emerge that can reliably forecast seismic data, perform state reconstruction, and interpolate across diverse parameter regimes.

## 4 LIMITATIONS & FUTURE WORK

The globally propagating seismic wavefield datasets used here are generated from an axisymmetric Earth model (van Driel et al., 2015), which captures the increase of seismic velocities and densities with depth but omits the heterogeneous geological structures that drive real-world geodynamics. Future datasets, such as those simulated from the REVEAL project (Thrastarson et al., 2024), or others generated for other planets (e.g., Mars - Stähler et al., 2021), could expand the parameter space and enable a more rigorous assessment of model generalization. Furthermore, distributed acoustic sensing (DAS) recordings with earthquake wavefields as more dominant features would add another layer of complexity. Such data contain higher frequency wavefields that exhibit extreme

scattering due to real, near-surface structure heterogeneity (e.g., features highlighted in Shi et al., 2025), challenging AI models to capture earthquake ground motion with societal relevance for natural hazard mitigation. Additional datasets may also come from laboratory experiments of earthquake behavior, which exhibit a particularly complex spatio-temporal pattern relevant to dynamical system studies (e.g., Corbi et al., 2022; 2025). Incorporating such experimental data would broaden the scope of our CTF, increasing the relevance of the models for broader tectonic implications and for scientists conducting laboratory experiments.

Additional limitations are inherent to our provided evaluations. While many models were tested in this work, there are many more that should be implemented and scored in the future. We hope that the Kaggle launch will inspire the scientific community to test many more models than what we have here, with the hope that community-driven engagement results in model improvements that vastly outperform our tested models across all tasks. There is also room for improvement in the tasks provided. While we start the CTF with a set of twelve measurements on the global seismic wavefields dataset, more tasks can be provided, yielding a deeper understanding of a model's capabilities. Finally, for the Kaggle competition, we will add a larger number of training data sets to potentially unlock the approximation capabilities of the models. With significantly more training data, there is the potential to improve the overall performance of many of the methods and beat the zero baseline. For instance, we will provide $P = 100$ simulations with varying initial data and parametrizations with the goal of predicting new initial data ($Q = 10$) with different parameter settings.

**Input:** $\mathbf{X}_{Jtrain} \in \mathbb{R}^{4m \times n}$ for $J = 1, 2, \cdots, P$;

**Output:** $\mathbf{X}_{Jpred} \in \mathbb{R}^{m \times n}$ for $J = 1, 2, \cdots, Q$;   **Scores:** $E_{13} - E_{22}$.

## 5 CONCLUSION

CTFs have been critical to the tremendous advancements in the big three ML fields of computer vision, natural language processing, and reinforcement learning. They have been proven to be catalysts for key model advancements by providing a clear objective measurement in a competitive environment. This work marks the beginning of incubating a similar environment in seismology by providing a challenging dataset and an objective quantification of model performance on tasks that are notoriously challenging. Our aim is to use our platform to evaluate the current state of methods and their usefulness in seismology as well as fostering an environment where novel methods are developed that excel for specific problem classes.

In this work, we introduce the Seismic Wavefield CTF, which quantifies the performance of modeling approaches on an amalgam of diverse tasks in seismology. As a first step, we provide the challenging global seismic wavefields dataset and evaluate 18 different models on the data to understand the usefulness of current ML models in forecasting, state reconstruction, and parametric variability. Our work demonstrates that the current state of ML is far behind any meaningful use on challenging seismic datasets. We hope to inspire researchers and engineers to identify and develop models that will advance the current modeling capabilities of seismic wavefields, ultimately leading to more accurate and reliable tools for earthquake science and subsurface characterization. By establishing a CTF, we aim to shift the research focus from incremental improvements on simplified problems to substantive breakthroughs on complex, realistic challenges. The CTF is designed to be a living framework, with plans to expand the datasets and tasks to continuously push the boundaries of what is possible in computational seismology.

### REPRODUCIBILITY STATEMENT

In this work, we introduce the Seismic Wavefield CTF. With the goal of evaluating ML and AI algorithms on seismic wavefields, we provide a publicly available GitHub repository containing the training datasets, hyperparameter tuning scripts, visualization notebooks, and all relevant documentation needed to reproduce and extend our results.

### ETHICS STATEMENT

The datasets in this CTF do not contain private or sensitive information. We release this CTF to encourage rigorous, reproducible research and urge the community to use it responsibly. To our best

understanding, the datasets and the source code does not cause harm to the scientific and global communities. Throughout this work, we adhered to the ICLR Code of Ethics.

ACKNOWLEDGMENTS

The authors acknowledge support from the National Science Foundation AI Institute in Dynamic Systems (grant number 2112085). This work is also partially supported by the Seismic Computational Platform for Empowering Discovery (SCOPED) project under the National Science Foundation (award numbers OAC-2103701). GM acknowledges support from the EPSRC programme grant in 'The Mathematics of Deep Learning' (project EP/V026259/1). AY is supported by the NSF Graduate Research Fellowship Program under Grant No. DGE-2140004. Any opinions, findings, and conclusions or recommendations expressed in this material are those of the author(s) and do not necessarily reflect the views of the sponsors.

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

# A APPENDIX

## A.1 DATASET FILES AND EVALUATION METRICS

The dataset files and evaluation metrics are provided in Table 2 and Table 3.

Table 2: Files and corresponding evaluation metrics ($E_1$–$E_{12}$) for benchmark datasets.

| Score | Test | Task | Train / Burn-in File(s) | Ground Truth File |
|---|---|---|---|---|
| $E_1$ | Forecasting | Short-time | $\mathbf{X}_{1\text{train}}$ | $\mathbf{X}_{1\text{test}}$ |
| $E_2$ | Forecasting | Long-time | $\mathbf{X}_{1\text{train}}$ | $\mathbf{X}_{1\text{test}}$ |
| $E_3$ | Noisy (medium) | Reconstruction (denoising) | $\mathbf{X}_{2\text{train}}$ | $\mathbf{X}_{2\text{test}}$ |
| $E_4$ | Noisy (medium) | Forecast (long-time) | $\mathbf{X}_{2\text{train}}$ | $\mathbf{X}_{3\text{test}}$ |
| $E_5$ | Noisy (high) | Reconstruction (denoising) | $\mathbf{X}_{3\text{train}}$ | $\mathbf{X}_{4\text{test}}$ |
| $E_6$ | Noisy (high) | Forecast (long-time) | $\mathbf{X}_{3\text{train}}$ | $\mathbf{X}_{5\text{test}}$ |
| $E_7$ | Limited Data (clean) | Forecast (short-time) | $\mathbf{X}_{4\text{train}}$ | $\mathbf{X}_{6\text{test}}$ |
| $E_8$ | Limited Data (clean) | Forecast (long-time) | $\mathbf{X}_{4\text{train}}$ | $\mathbf{X}_{6\text{test}}$ |
| $E_9$ | Limited Data (noisy) | Forecast (short-time) | $\mathbf{X}_{5\text{train}}$ | $\mathbf{X}_{7\text{test}}$ |
| $E_{10}$ | Limited Data (noisy) | Forecast (long-time) | $\mathbf{X}_{5\text{train}}$ | $\mathbf{X}_{7\text{test}}$ |
| $E_{11}$ | Parametric Generalization | Interpolation forecast | $\mathbf{X}_{6,7,8\text{train}}$ / $\mathbf{X}_{9\text{train}}$ | $\mathbf{X}_{8\text{test}}$ |
| $E_{12}$ | Parametric Generalization | Extrapolation forecast | $\mathbf{X}_{6,7,8\text{train}}$ / $\mathbf{X}_{10\text{train}}$ | $\mathbf{X}_{9\text{test}}$ |

Table 3: Matrix shapes and indices for the global wavefields dataset (left), DAS dataset (center), and 3D crustal wavefield dataset (right). Start and end index refer to relative time-steps in the simulation used to generate the dataset matrices. Each successive index represents one $\Delta t$ time-step. Matrix shapes follow the [Timesteps, Dimension] format.

| Matrix | Global Wavefields Dataset Shape | Start Index | End Index | Distributed Acoustic Sensing Dataset Shape | Start Index | End Index | 3D Crustal Wavefield Dataset Shape | Start Index | End Index |
|---|---|---|---|---|---|---|---|---|---|
| $\mathbf{X}_{1\text{train}}$ | [2000, 2048] | 0 | 2000 | [2000, 3000] | 0 | 2000 | [500, 62451] | 0 | 500 |
| $\mathbf{X}_{2\text{train}}$ | [2000, 2048] | 0 | 2000 | [2000, 3000] | 0 | 2000 | [500, 62451] | 0 | 500 |
| $\mathbf{X}_{3\text{train}}$ | [2000, 2048] | 0 | 2000 | [2000, 3000] | 0 | 2000 | [500, 62451] | 0 | 500 |
| $\mathbf{X}_{4\text{train}}$ | [500, 2048] | 1500 | 2000 | [500, 3000] | 1500 | 2000 | [200, 62451] | 300 | 500 |
| $\mathbf{X}_{5\text{train}}$ | [500, 2048] | 1500 | 2000 | [500, 3000] | 1500 | 2000 | [200, 62451] | 300 | 500 |
| $\mathbf{X}_{6\text{train}}$ | [2000, 2048] | 0 | 2000 | [2000, 3000] | 0 | 2000 | [500, 26508] | 0 | 500 |
| $\mathbf{X}_{7\text{train}}$ | [2000, 2048] | 0 | 2000 | [2000, 3000] | 0 | 2000 | [500, 26508] | 0 | 500 |
| $\mathbf{X}_{8\text{train}}$ | [2000, 2048] | 0 | 2000 | [2000, 3000] | 0 | 2000 | [500, 26508] | 0 | 500 |
| $\mathbf{X}_{9\text{train}}$ | [500, 2048] | 1500 | 2000 | [500, 3000] | 1500 | 2000 | [200, 26508] | 300 | 500 |
| $\mathbf{X}_{10\text{train}}$ | [500, 2048] | 1500 | 2000 | [500, 3000] | 1500 | 2000 | [200, 26508] | 300 | 500 |
| $\mathbf{X}_{1\text{test}}$ | [1000, 2048] | 2000 | 3000 | [1000, 3000] | 2000 | 3000 | [100, 62451] | 500 | 600 |
| $\mathbf{X}_{2\text{test}}$ | [2000, 2048] | 0 | 2000 | [2000, 3000] | 0 | 2000 | [500, 62451] | 0 | 500 |
| $\mathbf{X}_{3\text{test}}$ | [1000, 2048] | 2000 | 3000 | [1000, 3000] | 2000 | 3000 | [100, 62451] | 500 | 600 |
| $\mathbf{X}_{4\text{test}}$ | [2000, 2048] | 0 | 2000 | [2000, 3000] | 0 | 2000 | [500, 62451] | 0 | 500 |
| $\mathbf{X}_{5\text{test}}$ | [1000, 2048] | 2000 | 3000 | [1000, 3000] | 2000 | 3000 | [100, 62451] | 500 | 600 |
| $\mathbf{X}_{6\text{test}}$ | [1000, 2048] | 2000 | 3000 | [1000, 3000] | 2000 | 3000 | [100, 62451] | 500 | 600 |
| $\mathbf{X}_{7\text{test}}$ | [1000, 2048] | 2000 | 3000 | [1000, 3000] | 2000 | 3000 | [100, 62451] | 500 | 600 |
| $\mathbf{X}_{8\text{test}}$ | [1000, 2048] | 2000 | 3000 | [1000, 3000] | 2000 | 3000 | [100, 26508] | 500 | 600 |
| $\mathbf{X}_{9\text{test}}$ | [1000, 2048] | 2000 | 3000 | [1000, 3000] | 2000 | 3000 | [100, 26508] | 500 | 600 |

## A.2 EVALUATIONS

### A.2.1 HYPERPARAMETER OPTIMIZATION

Hyperparameter optimization is performed using our publicly available GitHub repository using the `tune_module.py` script. We employ Ray Tune Liaw et al. (2018) for systematic hyperparameter optimization across all models. Hyperparameters are defined in YAML configuration files specifying parameter types, bounds, and sampling distributions. Multiple parameter types are supported, including continuous distributions (uniform, log-uniform), discrete distributions (random integer, log-random integer), and categorical choices.

The optimization follows a trial-based approach where each trial randomly samples a hyperparameter configuration from the defined search space. Each trial trains the model using a train/validation split of the original training dataset. The `tune_module.py` script splits the training data into train and validation sets, using the latter exclusively for evaluation. Thus, the test set remains unseen during hyperparameter tuning.

For hyperparameter tuning metrics $E_1$ through $E_{10}$ we split the training matrix into two parts: the first 80% is the training data and the last 20% is the testing data. For hyperparameter tuning metric

$E_{11}$ we use matrices $\mathbf{X}_{6,8\text{train}}$ for the training data and $\mathbf{X}_{7\text{train}}$ for the testing data, and the burn-in matrix comes from $\mathbf{X}_{7\text{train}}$ as well. For hyperparameter tuning metric $E_{12}$ we use matrices $\mathbf{X}_{6,7\text{train}}$ for the training data and $\mathbf{X}_{8\text{train}}$ for the testing data, and the burn-in matrix comes from $\mathbf{X}_{8\text{train}}$ as well.

Optimization terminates when either a predefined number of trials or a time budget is reached. We employ ASHA (Asynchronous Successive Halving Algorithm) scheduling Li et al. (2020) for early stopping of poorly performing trials. Resource allocation is automatically managed, distributing trials across available computational resources.

For our results, each combination of model, dataset, and pair_id is allocated 8 hours of tuning time on dedicated nodes equipped with 1 NVIDIA A100 GPU with 40 GiB VRAM and 18 CPU cores from an Intel Xeon Platinum 8360Y processor with 120GiB RAM. Some models complete tuning in less than the allocated time.

### A.3 Models

#### A.3.1 Baselines

We implement two baseline models. One of the baselines predicts all zeros. The other baseline predicts the average of the input data per spatial dimension. We do not perform hyperparameter optimization for either of these models.

#### A.3.2 LSTM/ODE-LSTM

LSTM networks are a specialized type of recurrent neural network (RNN) designed to address the vanishing gradient problem inherent in traditional RNNs Hochreiter & Schmidhuber (1997). They achieve this through a unique architecture featuring memory cells and gating mechanisms (input, forget, and output gates), which regulate the flow of information over time. These gates enable LSTMs to selectively retain or discard historical data, making them particularly adept at capturing long-term dependencies in sequential data. In time-series forecasting, LSTMs excel at modeling temporal patterns, such as trends, seasonality, and irregular fluctuations, by leveraging past observations to predict future values. Their ability to handle complex, non-linear relationships and variable-length input sequences makes them a robust choice for tasks like stock prediction, energy load forecasting, or weather modeling, where historical context is critical to accurate predictions.

ODE-LSTMs are a flavor of LSTMs that try to further tackle the vanishing gradient problem by using an ODE solver to model the hidden state of the LSTM Coelho et al. (2024). They show that traditional LSTMs can still suffer from a vanishing or exploding gradient and provide theory demonstrating ODE-LSTMs do not suffer from either of these problems.

We evaluate both a classical LSTM and the ODE-LSTM by searching over the following hyperparameters: hidden_state_size (dimension of the latent space), seq_length (input sequence length), and lr (learning rate).

| hyperparameter | type | min (or options) | max (or none) |
|---|---|---|---|
| hidden_state_size | randint | 8 | 256 |
| seq_length | randint | 5 | 512 |
| lr | log_uniform | $10^{-5}$ | $10^{-2}$ |

Table 4: Hyperparameter search space for the ODE-LSTM and LSTM models on metrics $E_1$ through $E_6$ for the tested datasets. We train with a batch size of 128 for 200 epochs.

#### A.3.3 SpaceTime

State-Space Models (SSMs) are mathematical frameworks that describe systems using latent (hidden) states evolving over time, observed through measurable outputs. They are widely used in control theory, signal processing, and time-series analysis to model dynamic systems. Modern adaptations like S4 (Structured State Space for Sequence Modeling) and SpaceTime are deep learning variants of SSMs tailored for sequential data. These models parameterize state transitions with structured

| hyperparameter | type | min (or options) | max (or none) |
| --- | --- | --- | --- |
| hidden_state_size | randint | 8 | 256 |
| seq_length | randint | 5 | 74 |
| lr | log_uniform | $10^{-5}$ | $10^{-2}$ |

Table 5: Hyperparameter search space for the ODE-LSTM and LSTM models on metrics $E_7$ through $E_{12}$ for the tested datasets. We train with a batch size of 5 for $E_7$ through $E_{10}$ and a batch size of 128 for $E_{11}$ and $E_{12}$ for 200 epochs.

matrices to efficiently capture long-range dependencies while remaining computationally tractable. Unlike LSTMs, SSMs are particularly effective at time-series forecasting of long-range dependencies with minimal memory overhead.

SpaceTime Zhang et al. (2023) is one such SSM that claims to be a state-of-the-art model on time-series forecasting and classification tasks. The authors claim that their model captures "complex, long-range, and *autoregressive*" dependencies, can forecast over long horizons, and is efficient during training and inference. They demonstrate improved performance over the popular S4 SSM and NLinear.

Based on the hyperparameter optimization described in the original paper and the hyperparameters which can be adjusted in the publicly available code, we do a hyperparameter search over the following values: lag (input sequence length), horizon (output sequence length), n_blocks (number of SpaceTime layers in the model encoder), dropout, weight_decay, kernel_dim (dimension of SSM kernel in each block), and lr (learning rate).

| hyperparameter | type | min (or options) | max (or none) |
| --- | --- | --- | --- |
| lag | randint | 32 | 512 |
| horizon | randint | 32 | 512 |
| n_blocks | choice | {3,4,5,6} | · |
| dropout | choice | {0, 0.25} | · |
| weight_decay | choice | {0, 0.0001} | · |
| kernel_dim | choice | {32,64,128} | · |
| lr | log_uniform | $10^{-5}$ | $10^{-2}$ |

Table 6: Hyperparameter search space for the SpaceTime model on metrics $E_1$ through $E_6$ for the tested datasets. We train with a batch size of 128 for 200 epochs.

| hyperparameter | type | min (or options) | max (or none) |
| --- | --- | --- | --- |
| lag | randint | 10 | 45 |
| horizon | randint | 10 | 45 |
| n_blocks | choice | {3,4,5,6} | · |
| dropout | choice | {0, 0.25} | · |
| weight_decay | choice | {0, 0.0001} | · |
| kernel_dim | choice | {32,64,128} | · |
| lr | log_uniform | $10^{-5}$ | $10^{-2}$ |

Table 7: Hyperparameter search space for the SpaceTime model on metrics $E_7$ through $E_{10}$ for the tested datasets. We train with a batch size of 5 for 200 epochs.

### A.3.4 DEEP OPERATOR NETWORKS

Deep Operator Networks (DeepONets) Lu et al. (2021a) recently emerged as a powerful tool designed to efficiently model high-dimensional physical systems and complex input-output relationships, as well as to solve challenging problems in scientific machine learning and engineering, such as partial differential equations. Specifically, DeepONets are a class of neural operators which decompose an operator $G : \mathcal{V} \to \mathcal{U}$ between infinite-dimensional functional spaces $\mathcal{V}$ and $\mathcal{U}$ into two cooperating sub-networks, namely *branch* and *trunk net*. The trunk encodes the input function

| hyperparameter | type | min (or options) | max (or none) |
|---|---|---|---|
| lag | randint | 10 | 45 |
| horizon | randint | 10 | 45 |
| n_blocks | choice | {3,4,5,6} | . |
| dropout | choice | {0, 0.25} | . |
| weight_decay | choice | {0, 0.0001} | . |
| kernel_dim | choice | {32,64,128} | . |
| lr | log_uniform | $10^{-5}$ | $10^{-2}$ |

Table 8: Hyperparameter search space for the SpaceTime model on metrics $E_{11}$ through $E_{12}$ for the tested datasets. We train with a batch size of 128 for 200 epochs.

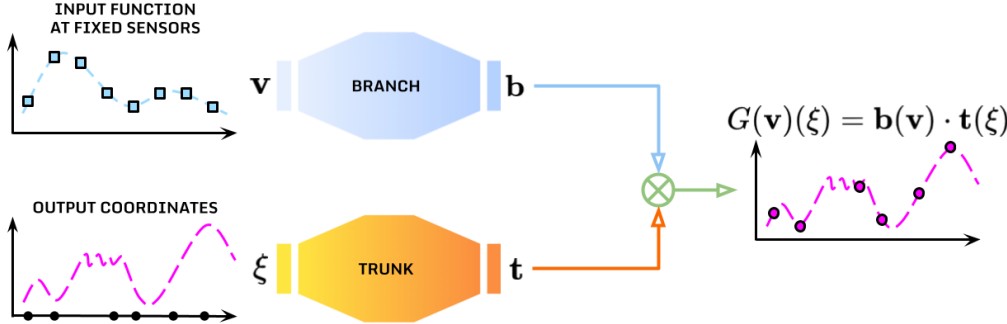

Figure 3: Architecture of the Deep Operator Network. The target field at the evaluation point $\xi$ is approximated by the inner product of the outputs of the branch net, which takes as input the measurements $\mathbf{v}$ of the input function $v \in \mathcal{V}$ and returns a set of coefficients $\mathbf{b}(\mathbf{v})$, and the trunk net, which encodes the coordinates $\xi$ into a vector $\mathbf{t}(\xi)$.

$v \in \mathcal{V} : \Omega' \subset \mathbb{R}^d \to \mathbb{R}^{n_v}$ – which is typically sampled at a finite set of $n$ fixed sensors, resulting in the measurement vector $\mathbf{v} \in \mathbb{R}^{n \cdot n_v}$ – into $p$ coefficients $\mathbf{b}(v) \in \mathbb{R}^p$. Instead, the branch net provides the evaluation of a neural learnable $p$-dimensional basis $\mathbf{t}(\xi) \in \mathbb{R}^p$ at the spatial coordinates $\xi$ in the domain $\Omega \subset \mathbb{R}^d$. Doing so, the value of the output function $u \in \mathcal{U} : \Omega \to \mathbb{R}^{n_u}$ at the evaluation point $\xi \in \Omega$ is approximated through the basis expansion

$$u(\xi) = G(v)(\xi) \approx \mathbf{b}(v) \cdot \mathbf{t}(\xi).$$

See Lu et al. (2021a); Chen & Chen (1995); Lu et al. (2022) for a complete presentation of Deep-ONets, including also universal approximation theorems for operators. A graphical summary of the DeepONet architecture is available in Figure 3.

**DeepONets for dynamical systems**   DeepONets are versatile neural architectures designed to learn mappings between functional spaces. DeepONets are traditionally exploited for inferring the space-time evolution of physical variables, such as the solution of partial differential equations, starting from known quantities, such as forcing terms, initial conditions, parameters or control variables Lu et al. (2021a; 2022); Wang et al. (2021); Kontolati et al. (2024). However, it is possible to adapt and employ DeepONets in the proposed CTF in order to model and forecast time-series data and dynamical systems, as proposed by, e.g., Chen et al. (2024a;b); Lin et al. (2023); He et al. (2024a;b); Michałowska et al. (2024). Specifically, we consider the operator

$$u_t(\xi) = G(u_{t-1}, ..., u_{t-k})(\xi) \approx \mathbf{b}(\mathbf{u}_{t-1}, ..., \mathbf{u}_{t-k}) \cdot \mathbf{t}(\xi)$$

where $u_t : \Omega \to \mathbb{R}^{n_u}$ and $\mathbf{u}_t \in \mathbb{R}^n$ are, respectively, the solution of the dynamical system under investigation at time $t$ and the corresponding spatial discretization, $k$ is the lag parameter and $\xi \in \Omega \subset \mathbb{R}^d$ are the spatial coordinates where to predict the evolution of the dynamics. Along with the evaluation point $\xi$, the trunk input may be enlarged with the time instance $t$ or the time-step $\Delta t$, as proposed by Lu et al. (2022); Lin et al. (2023).

**DeepONets implementation** The implementation of DeepONets within the proposed CTF is based on the *DeepXDE* library Lu et al. (2021b). In particular, when dealing with forecasting tasks, we predict the state evolution in an autoregressive manner, and we enlarge the trunk input with the time-step $\Delta t$, as it results in better performance. As proposed by Lu et al. (2022), we consider a scaler to normalize the data before training. Moreover, we employ branch and trunk networks with the same number of neurons per hidden layer, so as to reduce the number of hyperparameters.

The tested datasets deal with one-dimensional scalar-valued functions, that is $d = n_u = 1$. The datasets are discretized and evaluated over uniformly spaced domains. Notice that we take into account the same locations across all the input-output pairs, resulting in a lower computational cost.

**Hyperparameters** The DeepONet hyperparameters mainly concern the neural network architectures and the corresponding training procedure. In addition, the lag parameter determines the length of the past state history fed into the branch input for forecasting. Notice that the lag value cannot be larger than the dimension of burn-in data, and it is set equal to zero when dealing with reconstruction tasks. Table 9 provides a summary of the hyperparameters in play, along with the corresponding search spaces explored for hyperparameters tuning.

| hyperparameter | type | min (or options) | max (or none) |
|---|---|---|---|
| lag | integer | 1 | 99 |
| branch_layers | integer | 1 | 5 |
| trunk_layers | integer | 1 | 5 |
| neurons | integer | 1 | 512 |
| activation | choice | {"tanh", "relu", "elu"} | . |
| initialization | choice | {"Glorot normal", "He normal"} | . |
| optimizer | choice | { "adam", "L-BFGS" } | . |
| learning_rate | loguniform | $10^{-5}$ | $10^{-1}$ |
| epochs | integer | 10000 | 10000 |

Table 9: Hyperparameter search space for DeepONet.

### A.3.5 SPARSE IDENTIFICATION OF NONLINEAR DYNAMICS

Sparse Identification of Nonlinear Dynamics (SINDy) Brunton et al. (2016) is a powerful algorithm designed to discover interpretable and parsimonious governing equations from time-series data. Given the data matrices

$$X = \begin{bmatrix} x_1(t_1) & x_1(t_2) & ... & x_1(t_m) \\ \vdots & \vdots & \ddots & \vdots \\ x_n(t_1) & x_n(t_2) & ... & x_n(t_m) \end{bmatrix}; \quad \dot{X} = \begin{bmatrix} \dot{x}_1(t_1) & \dot{x}_1(t_2) & ... & \dot{x}_1(t_m) \\ \vdots & \vdots & \ddots & \vdots \\ \dot{x}_n(t_1) & \dot{x}_n(t_2) & ... & \dot{x}_n(t_m) \end{bmatrix}$$

collecting, respectively, the state vector $\mathbf{x}(t) = [x_1(t), ..., x_n(t)]$ and the corresponding time derivatives $\dot{\mathbf{x}}(t) = [\dot{x}_1(t), ..., \dot{x}_n(t)]$ at the time instances $t_1, ..., t_m$, we aim at identifying the (possibly nonlinear) underlying governing equation $\dot{\mathbf{x}}(t) = f(\mathbf{x}(t))$. To this aim, SINDy considers the following approximation

$$\dot{X} = \Theta(X)\Xi$$

where $\Theta(X)$ is a library of candidate regression terms, such as polynomials or trigonometric functions, while $\Xi$ are the corresponding regression coefficients. Sparsity promoting strategies are crucial to identify simple and interpretable dynamical systems, capable of avoiding overfitting and accurately extrapolating beyond training data. In particular, the regression coefficients $\Xi$ are determined through sparse regression strategies, such as Least Absolute Shrinkage and Selection Operator (LASSO) or Sequentially Thresholded Least SQuares (STLSQ). See Figure 4 for a scheme of the SINDy algorithm on the Lorenz system.

SINDy can easily handle parametric dependencies: indeed, augmenting the state vector with the (possibly time-dependent) parameter values $\boldsymbol{\mu}$ and adding $\boldsymbol{\mu}$-dependent terms in the library $\Theta(X, \boldsymbol{\mu})$, it is possible to identify parametric sparse dynamical systems.

Identifying sparse dynamical systems from high-dimensional data may be computationally expensive. A possible workaround is given by dimensionality reduction techniques, such as Proper Orthogonal

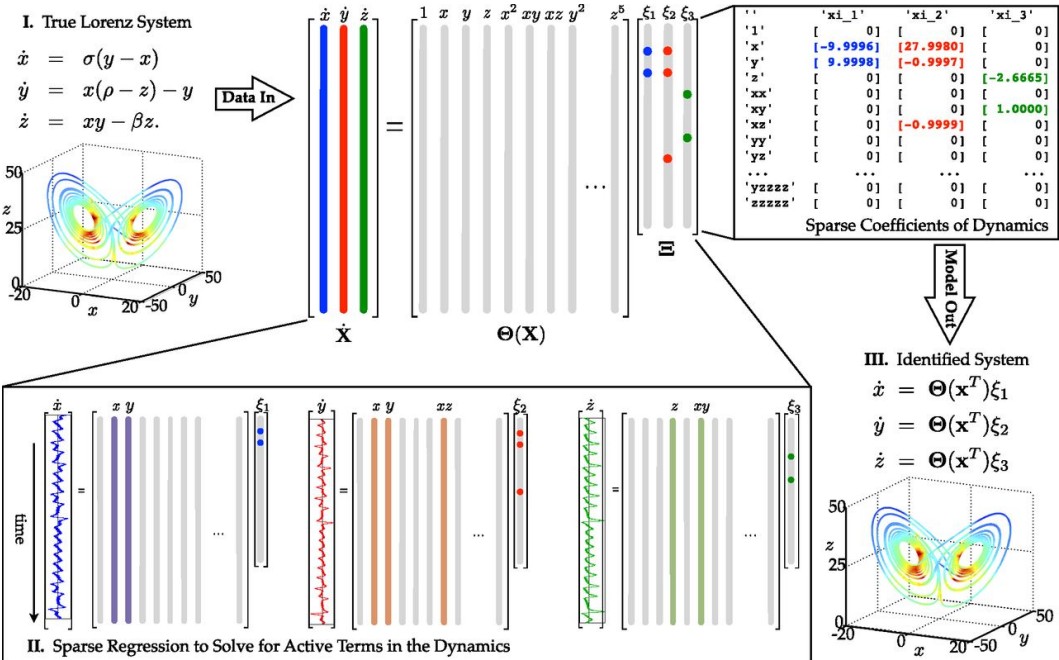

Figure 4: Schematic of the Sparse Identification of Nonlinear Dynamics (SINDy) algorithm from Brunton et al. (2016), demonstrated on the Lorenz equations. The temporal evolution of the state variable and its derivative are collected in the data matrices $X$ and $\dot{X}$. The dynamical system $\dot{X} = \Theta(X)\Xi$ is then identified through sparsity promoting algorithms.

Decomposition (POD) Brunton et al. (2016) or autoencoders Champion et al. (2019), which project state snapshots onto a low-dimensional manifold. SINDy can thus be applied on the low-dimensional latent variables, allowing for efficient and accurate forecasting of the high-dimensional state evolution.

**SINDy implementation**    The implementation of SINDy is based on the *PySINDy* library de Silva et al. (2020). After collecting the data and approximating the time derivatives through numerical schemes, the SINDy algorithm is applied to identify a sparse dynamical system describing the data evolution over time. The integrator *solve_ivp* by *scipy* Virtanen et al. (2020) is considered to simulate the system and to predict future state values. Notice that, whenever the identified model is very complex and the integrator fails, the static dynamical system $\dot{\mathbf{x}} = 0$ is employed.

Proper Orthogonal Decomposition (POD) is thus exploited to compress the temporal data of the seismic wavefields data, and SINDy is applied to identify the dynamics of the most energetic POD coefficients. Therefore, the predictions are retrieved by integrating the SINDy model and projecting the POD coefficients onto the original high-dimensional state space.

Parametric SINDy models are considered when testing the ability of the model to generalize to different parameter values. Since the parameter values employed for data generation are not publicly available, we take into account fictitious values mimicking the interpolatory and extrapolatory regimes.

**Hyperparameters**    The SINDy algorithm can exploit different differentiation methods to approximate time derivatives, different terms in the library $\Theta(X)$ – such as, e.g., polynomials and/or trigonometric functions up to a chosen order – as well as different sparse regression techniques. Table 10 provides a summary of the hyperparameters in play, along with the corresponding search spaces explored for hyperparameter tuning.

| hyperparameter | type | min (or options) | max (or none) |
|---|---|---|---|
| POD_modes | integer | 1 | 50 |
| differentiation_method | choice | { "finite_difference", "spline", "savitzky_golay", "spectral", "trend_filtered", "kalman" } | . |
| differentiation_method_order | integer | 1 | 10 |
| feature_library | choice | { "polynomial", "Fourier", "mixed" } | . |
| feature_library_order | integer | 1 | 10 |
| optimizer | choice | {"STLSQ", "SR3", "SSR", "FROLS"} | . |
| threshold | choice | { "adam", "L-BFGS" } | . |
| learning_rate | loguniform | $10^{-3}$ | $10^3$ |
| alpha | loguniform | $10^{-3}$ | $10^1$ |

Table 10: Hyperparameter search space for SINDy.

### A.3.6 DYNAMIC MODE DECOMPOSITION

The Dynamic Mode Decomposition (DMD) is a data-driven method developed by Schmid (2010) in the fluid dynamics community to identify spatio-temporal coherent structures from high-dimensional data. The DMD algorithm is based on the Singular Value Decomposition (SVD) of a data matrix; in particular, DMD is able to provide a modal decomposition where each mode consists of spatially correlated structures that have the same linear behavior in time. The DMD method is found to have a significant connection with the Koopman operator theory Rowley et al. (2009): in particular, the DMD can be formulated as an algorithm able to learn the best-fit linear dynamical system to advance in time (Figure 5).

There are many variants of DMD, connected to existing techniques from system identification and modal extraction Brunton & Kutz (2022). Here, we will provide a brief overview of the underlying idea of the original DMD algorithm, from which all the other variants can be derived. The first step is to collect a set of snapshots of the system at different time steps. The data matrix is then constructed by stacking the snapshots in columns, i.e., $\mathbf{X} = [\mathbf{x}_1, \mathbf{x}_2, \ldots, \mathbf{x}_{N_t}] \in \mathbb{C}^{\mathcal{N}_h \times N_t}$, where $\mathbf{x}_k \in \mathbb{C}^{\mathcal{N}_h}$ is the $k$-th snapshot at time $t_k$ and $N_t$ is the number of snapshots. The original formulation from Schmid (2010); Rowley et al. (2009) supposed uniform sampling in time, i.e. $t_k = k\Delta t$, where $\Delta t$ is the time step and $t_{k+1} = t_k + \Delta t$. Overall, the DMD algorithm seeks the leading spectral decomposition of the best-fit linear operator $\mathbb{A} \in \mathbb{C}^{\mathcal{N}_h \times \mathcal{N}_h}$ that advances the system in time, i.e.

$$\mathbf{x}_{k+1} \approx \mathbb{A}\mathbf{x}_k \quad \longleftrightarrow \quad \mathbf{X}_{[2:N_t]} \approx \mathbb{A}\mathbf{X}_{[1:N_t-1]}$$

As we said above, the DMD algorithm is based on the SVD of the data matrix $\mathbf{X}$ of rank $r$, which can be written as $\mathbb{X} \simeq \mathbf{U}\mathbf{\Sigma}\mathbf{V}^*$: $\mathbf{U} \in \mathbb{C}^{\mathcal{N}_h \times r}$ represents the left singular vectors and are also known as modes, describing the dominant spatial structures extracted from the data; the diagonal matrix $\mathbf{\Sigma} \in \mathbb{R}^{r \times r}$ contains the singular values, which are related to the energy/information retained by the modes; in the end, $\mathbf{V}^* \in \mathbb{C}^{r \times N_t}$ represents the right singular vectors, which are related to the temporal dynamics of the modes. This compression operation allows to compute the dynamical matrix $\mathbb{A}$ in a more efficient way J. Nathan Kutz et al. (2016); Brunton & Kutz (2022), avoiding the direct inversion of the high-dimensional snapshot matrix.

Indeed, in the literature different variants of DMD have been proposed: in this context, the High-Order DMD (HODMD) Le Clainche & Vega (2017b), which exploits time delay embedding to fit the optimal Koopman Operator, and the Optimised DMD (OptDMD) Askham & Kutz (2018b); Sashidhar & Kutz (2022), which is a variant of DMD that can also use the Bagging algorithm to improve the robustness of the DMD algorithm against noise. This latter variant has been shown to be the most robust and stable algorithm for real-world applications Faraji et al. (2024). The implementation of the DMD algorithm is available in the `pyDMD` package Demo et al. (2018); Ichinaga et al. (2024), which is a Python library for DMD and its variants. The library is designed to be easy to use and flexible, allowing users to customise the algorithm for their specific needs.

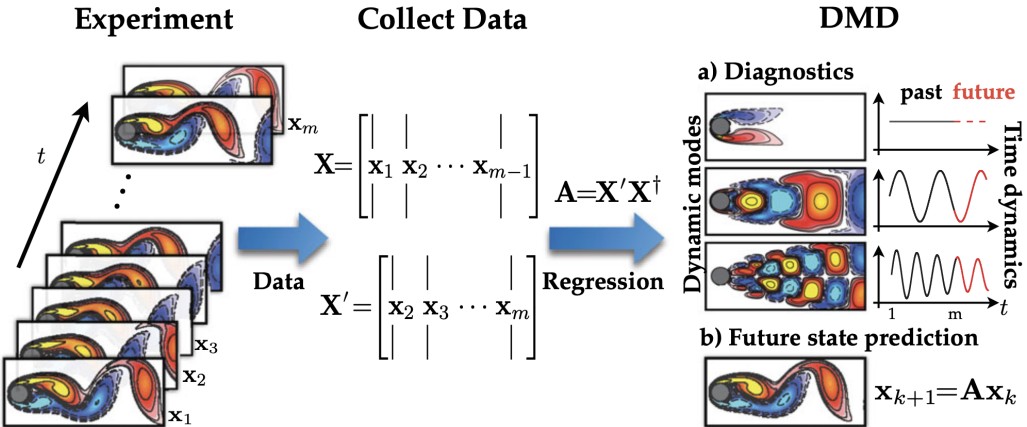

Figure 5: Scheme of the Dynamic Mode Decomposition algorithm from J. Nathan Kutz et al. (2016). The data matrix $\mathbf{X}$ is constructed by stacking the snapshots in columns. The SVD of the data matrix is computed, and the dynamical matrix is fitted to the data. This allows us to compute the state of the system for future time instances.

**Parametric DMD**   The extension of DMD to parametric systems is a recent development in the field of system identification. Different approaches have been proposed in the literature; in this work, the implementation of Andreuzzi et al. Andreuzzi et al. (2023b) within `pyDMD` is adopted. Up to now, the package does not support the OptDMD algorithm directly, we have implemented a wrapper to use the OptDMD algorithm with the parametric DMD following the same approach of the package, based on the interpolation of the forecasted reduced dynamics. We appreciate that further work and rigorous testing of this implementation are planned for future work. Similar to SINDy, since the parameter values employed for data generation are not publicly available, fictitious values mimicking the interpolatory and extrapolatory regimes have been used.

**Hyperparameter tuning**   The hyperparameters of the DMD algorithm depend on the specific variant adopted. Every DMD algorithm has a set of hyperparameters that can be tuned to improve the performance of the algorithm; however, the rank of the SVD is common to all of them and plays a crucial role in the reduction process. The HODMD algorithm also includes the delay embedding, defining the size of the lagging window to use. The OptDMD algorithm can also put constraints on the DMD eigenvalues to ensure that the dynamics follow a certain behavior. In the end, the parametric DMD can operate in two different modes: partitioned and monolithic. The hyperparameters of both DMD algorithms are listed in Tables 11 and 12.

| hyperparameter | type | min (or options) | max (or none) |
|---|---|---|---|
| rank | randint | 20 | 200 |
| delay | randint | 0 | 50 |
| parametric | choice | {"partitioned", "monolithic"} | |

Table 11: Hyperparameter search space for the HODMD algorithm for the tested datasets (the parametric hyperparameter has an effect only for metrics $E_{11}$ and $E_{12}$).

### A.3.7   KOOPMAN OPERATOR-BASED DYNAMIC SYSTEM PREDICTION

**The Koopman operator**   Koopman operator theory is a useful tool that has found increasing attention in the data-driven scientific computing community and can essentially be seen as an extension of dynamic mode decomposition - viewing the state space of the dynamic system through the lens of nonlinear observables. This point-of-view dates back to early work by Koopman (1931); Koopman & von Neumann (1932) and a modern review can be found in Brunton et al. (2022). We

| hyperparameter | type | min (or options) | max (or none) |
|---|---|---|---|
| rank | randint | 50 | 200 |
| delay | randint | 0 | 4 |
| parametric | choice | { "partitioned", "monolithic"} | |
| eig_constraints | choice | { "none", "stable", "conjugate_pairs"} | |

Table 12: Hyperparameter search space for the OptDMD algorithm for the tested datasets (the parametric hyperparameter has an effect only for metrics $E_{11}$ and $E_{12}$).

outline the method briefly before describing the set-up for the chosen implementation and our testing on the CTF. Consider a dynamical system (either an ODE or a semi-discretisiation of a PDE) of the form:

$$\frac{d\mathbf{x}}{dt} = \mathbf{f}(\mathbf{x}),$$

where $\mathbf{f} : \mathbb{R}^N \to \mathbb{R}^N$ may be a nonlinear forcing. The central idea in Koopman operator theory is then to learn a coordinate transform (i.e. a set of nonlinear observables) $\Phi : \mathbb{R}^N \to \mathbb{R}^M$, under which the dynamics becomes (approximately) linear, i.e.

$$\frac{d\mathbf{z}}{dt} \approx \mathbf{A}\mathbf{z}, \quad \mathbf{z}(t) = \Phi(\mathbf{x}(t)).$$

In this new coordinate system, the exact solution of the linear dynamics is straightforward. The inference of $\Phi$ and $\mathbf{A}$ can be formulated as a regression problem.

**Numerical implementation and parameter choices** In the Seismic Wavefield CTF we use the `PyKoopman` Python library as the main reference point for the Koopman method for dynamic system prediction Pan et al. (2024). The Python package serves as a good reference since it is regularly maintained and has an up-to-date implementation of several central features of the Koopman operator framework. As mentioned above there are two central parameters that affect the performance of the Koopman method: the observables and the regression method. Exploiting the existing implementation in `PyKoopman` we allowed in our CTF testing the variation of the following set of parameters:

- Type of observable: Options include the identity, polynomials of variable degree, time delay (of variable depth), radial basis functions (of variable number) and random Fourier features, as well as the concatenation of all of the aforementioned observables with the identity;

- Type of regressor: DMD, EDMD, HAVOK and KDMD;

- Regressor rank;

- Least-squares regularisation and rank of the regularisation (this option is implemented only in EDMD and KDMD).

Note that in principle a neural network-based DMD is also implemented in the PyKoopman package, but in our fine-tuning we found that this lead consistently to worse performance than the above four types of regressors thus we excluded it from the hyperparameter tuning.

**Parametric PyKoopman** Out-of-the-box `PyKoopman` does not have a parametric implementation, thus in order to test the Koopman method on task (d), we loosely follow Andreuzzi et al. (2023a); Guo et al. (2025) and implement a custom parametric version of `PyKoopman` by spline interpolation of the learned Koopman operator and corresponding eigenfunctions. We acknowledge that further work and rigorous testing of various parametric versions of the Koopman method are required to identify the best performing implementation for task (d).

**Hyperparameter tuning** Based on the available choices implemented in the PyKoopman package and the examples described in the documentation Pan et al. (2023), we performed a hyperparameter search over the following parameters: type of observable and potential concatenation with the identity, observables integer parameter (representing the polynomial degree in case of polynomial observables, the number of time delay steps in the case of time delay observables and the parameter $D$ in the random

Fourier feature case), the number of centers for the radial basis function observables, observables float parameter (representing the radial basis function kernel width and the parameter $\gamma$ in the radial basis function case respectively), regressor type, regressor rank, TLSQ rank (the regularization rank called only when the regressor is EDMD and KDMD). The details of the parameter space explored are shown in Table 13.

| hyperparameter | type | min (or options) | max (or none) |
|---|---|---|---|
| observables | choice | {Identity, Polynomial, TimeDelay, RadialBasisFunctions, RandomFourierFeatures} | . |
| Identity concatenation | choice | {true, false} | |
| Integer parameter | randint | 1 | 10 |
| # RBF centers | randint | 10 | 1000 |
| Float parameter | uniform | 0.5 | 2.0 |
| regressor type | choice | {DMD,EDMD, HAVOK, KDMD} | . |
| regressor rank | randint | 1 | 200 |
| TLSQ rank | randint | 1 | 200 |

Table 13: Hyperparameter search space for the PyKoopman model.

### A.3.8 RESERVOIR COMPUTING

In its broadest sense, reservoir computing (RC) is a general machine learning framework for processing sequential data. RC functions by projecting data into a high-dimensional dynamical system and training a simple readout from these dynamics back to a quantity or signal of interest. Although there exists a large and ever-growing body of literature on leveraging physical systems to act as high-dimensional "reservoirs" Tanaka et al. (2019), the most common form of RC remains an echo state network (ESN) Jaeger (2001a); Maass et al. (2002). ESNs are a form of recurrent neural network (RNN) that have been demonstrated to achieve state-of-the-art performance in the forecasting of chaotic dynamical systems Platt et al. (2022); Vlachas et al. (2020). We now introduce the specific form of ESN we use in evaluating performance on the CTF datasets, following many of the conventions presented in Platt et al. (2022).

**ESNs for Low-Dimensional Systems.** Given a time series $u_0, \ldots, u_T$, a randomly instantiated, high-dimensional dynamical system is evolved according to

$$h_{t+1} = (1 - \alpha)h_t + \alpha \tanh\left(W_{hh}h_t + W_{hu}u_t + \sigma_b \mathbf{1}\right) \tag{4}$$

where $\alpha$ is the so-called leak rate hyperparameter, $W_{hh}$ and $W_{hu}$ are fixed, random matrices, $\sigma_b$ is a bias hyperparameter and $\mathbf{1}$ denotes a vector of ones. $W_{hh} \in \mathbb{R}^{N_h \times N_h}$ ($N_h$ denotes the number of entries in $h$) is taken to be a random, sparse matrix with density $\approx 2\%$ and non-zero entries sampled from $\mathcal{U}(-1, 1)$ and then scaled such that the spectral radius of $W_{hh}$ is $\rho$. $W_{hu} \in \mathbb{R}^{N_h \times N_u}$ ($N_u$ denotes the number of entries in $u$) is a random matrix with each entry drawn independently from $\mathcal{U}(-\sigma, \sigma)$. Initializing $h_0$ as $h_0 = \mathbf{0}$, we generate a sequence of training reservoir states $h_0, \ldots, h_T$. We discard the initial $N_{spin}$ training states as an initial transient and then perform a Ridge regression (with Tikhonov regularization $\beta$) to learn a linear map $W_{uh}$ such that

$$W_{uh}g(h_i) \approx u_i. \tag{5}$$

$g : N_h \to N_h$ is often taken to be the identity map or simply squaring every odd indexed entry of $h_i$. We assume the latter convention, following the work of Pathak et al Pathak et al. (2018). Once trained the reservoir dynamics can be run autonomously as

$$h_{t+1} = (1 - \alpha)h_t + \alpha \tanh\left(W_{hh}h_t + W_{hu}W_{uh}g(h_t) + \sigma_b \mathbf{1}\right) \tag{6}$$

to obtain a forecast of arbitrary length.

**ESNs for High-Dimensional Systems.** RC approaches typically rely on the latent dimension $N_h >> N_u$. However, the computational cost of the previous algorithm scales roughly quadratically with $N_h$. Thus, while the above approach works well for relatively small systems, without modification it does not scale well to large states such as those encountered in PDE simulations. Pathak et al.

introduced a parallel reservoir approach to address this issue by dividing a high-dimensional input into $g$ lower dimensional "chunks" Pathak et al. (2018). A single reservoir then accepts as input only $N_u/g + 2L$ values, where $L$ is a locality parameter that dictates the overlap of input for two adjacent reservoirs. The output of the single reservoir is only $g$ entries of the state. Since computational cost grows linearly in the number of reservoirs, this parallel approach allows for the application of RC to higher dimensional systems. Each individual reservoir is trained exactly as for Low-Dimensional systems; there are now just $g$ reservoirs representing different regions of the domain.

Since we introduce two new hyperparameters in the parallel setup ($L$ and $g$), when we perform our hyperparameter tuning for the the tested seismic wavefield systems we fix $\alpha = 1$ and $\sigma_b = 0$, following the work of Pathak et al. The complete hyperparameter search space for the tested datasets are given in Table 14.

| hyperparameter | type | min (or options) | max (or none) |
|:---:|:---:|:---:|:---:|
| $g$ | choice | $\{16, 32, 64, 128\}$ | . |
| $\sigma$ | loguniform | $0.0001$ | $1.0$ |
| $L$ | randint | $1$ | $10$ |
| $\rho$ | uniform | $0.02$ | $1$ |
| $\beta$ | loguniform | $10^{-10}$ | $10^{-1}$ |
| $N_h$ | randint | $500$ | $3000$ |

Table 14: Hyperparameter search space for the reservoir model on the tested datasets.

### A.3.9 FOURIER NEURAL OPERATOR

Neural operators are a class of machine learning models designed to learn mappings between function spaces, in contrast to the finite-dimensional Euclidean spaces typically used in conventional neural networks. Although the inputs and outputs are discretized in practice, neural operators aim to generalize across discretizations and treat functions as the primary objects of learning.

The Fourier Neural Operator (FNO), in particular, is a neural operator architecture that replaces the kernel integral operator with a convolution operator defined in Fourier space, which allows for learning of operators in the frequency domain. It maps the input to the frequency domain using the Fourier transform, applies spectral convolution by multiplying learnable weights with the lower Fourier modes, and maps the result back to the physical domain via the inverse Fourier transform. This allows the model to learn families of PDEs, rather than solving individual instances. Without the high cost of evaluating integral operators, it maintains competitive computational efficiency.

Let $D \subset \mathbb{R}^d$ be a bounded domain. We consider learning an operator $G$ that maps between function spaces:

$$G : \mathcal{A} \to \mathcal{U} \tag{7}$$

where $\mathcal{A} = L^2(D; \mathbb{R}^{d_a})$ is the input function space and $\mathcal{U} = L^2(D; \mathbb{R}^{d_u})$ is the output function space.

Given an input function $a \in \mathcal{A}$, the FNO approximates the operator $G$ through a kernel integral operator:

$$G(a)(x) = \sigma \left( W a(x) + b + \int_D \kappa(x, y) a(y) \, dy \right) \tag{8}$$

where $W \in \mathbb{R}^{d_u \times d_a}$ is a linear transformation, $b \in \mathbb{R}^{d_u}$ is a bias term, $\kappa : D \times D \to \mathbb{R}^{d_u \times d_a}$ is a learnable kernel function, and $\sigma : \mathbb{R}^{d_u} \to \mathbb{R}^{d_u}$ is a pointwise non-linear activation function.

The kernel is parameterized in Fourier space as:

$$\kappa(x, y) = \sum_{k \in \mathbb{Z}^d} \widehat{\kappa}(k) e^{2\pi i k \cdot (x-y)} \tag{9}$$

where $\widehat{\kappa}(k)$ are the Fourier coefficients of the kernel. The translation-invariant kernel $\kappa(x, y) = \kappa(x - y)$ enables efficient convolution. This leads to the implementation:

$$G(a)(x) = \sigma \left( W a(x) + b + \sum_{k \in \mathbb{Z}^d} \widehat{\kappa}(k) \widehat{a}(k) e^{2\pi i k \cdot x} \right) \tag{10}$$

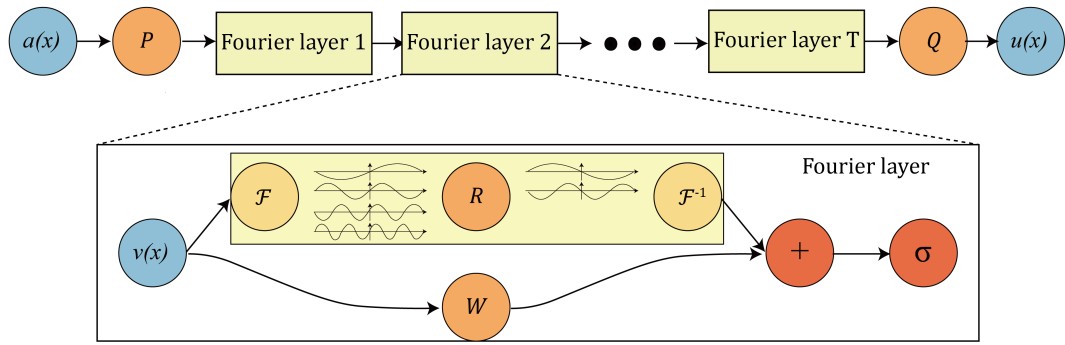

Figure 6: Architecture of the Fourier Neural Operator from Li et al. (2021)

where $\widehat{a}(k)$ represent the Fourier coefficients of the input function $a$. In practice, the sum over $k \in \mathbb{Z}^d$ is truncated to a finite number of low-frequency modes.

**Model Architecture** The architecture (Figure 6) begins with an initial fully connected multilayer perceptron (MLP) that projects the input to a higher-dimensional space, followed by four Fourier layers, and concludes with two fully connected MLPs that project the output to the desired dimensions.

Each Fourier layer performs a spectral convolution by first transforming the data into the frequency domain using Fast Fourier Transform (FFT), then multiplying the Fourier coefficients with learnable weights in the frequency space, and finally transforming back to physical space using inverse FFT. The Fourier layer only keeps a limited number of the lower Fourier modes, with high modes being filtered out. Additionally, each layer adds a linearly transformed version of its input to the output of the spectral convolution, which helps preserve local features and adds flexibility to the layer's expressiveness. Every Fourier layer is followed by a GELU activation function to introduce non-linearity.

**Hyperparameters** Based on our implementation of the FNO model, which closely follows that of the original paper, we test the hyperparameters as shown in Table 15. The number of Fourier modes is tuned separately for each mode.

| hyperparameter | type | range or options |
|---|---|---|
| Fourier modes | integer | [8,32] |
| Network width | integer | [32, 128] |
| Batch size | choice | 16, 32, 64, 128 |
| Learning rate (lr) | loguniform | [0.0001, 0.01] |

Table 15: Hyperparameter search space for the FNO model.

### A.3.10 KOLMOGOROV-ARNOLD NETWORKS

Kolmogorov–Arnold Networks (KANs) are a recently proposed alternative to traditional Multi-Layer Perceptrons (MLPs) Liu et al. (2024c). With learnable activation functions placed on edges that replace linear weights, KANs have been shown to provide improved accuracy and greater interpretability compared to traditional methods.
KANs were inspired by the Kolmogrov-Arnold representation theorem which posits that any multivariate continuous function $f$ on a bounded domain can be expressed as a finite composition and addition of univariate continuous functions Kolmogorov (1957). In other words, for a smooth function $f : [0,1]^n \to \mathbb{R}$,

$$f(\mathbf{x}) = f(x_1, x_2, ..., x_n) = \sum_{q=1}^{2n+1} \Phi_q \left( \sum_{p=1}^{n} \phi_{q,p}(x_p) \right) \tag{11}$$

where $\phi_{q,p} : [0,1] \to \mathbb{R}$ and $\Phi_q : \mathbb{R} \to \mathbb{R}$.

**Model Architecture** While the Kolmogrov-Arnold representation theorem is restricted to a small number of terms and only two hidden layers, this theorem can be generalized to increase the width and depth of the network. A single KAN layer is defined as a matrix of 1D functions thus the inner and outer functions in Equation 11, $\phi_{q,p}$ and $\Phi_q$, each represent a single KAN layer. A deeper network can be constructed by adding more KAN layers. A general KAN network with $L$ layers can be represented as a composition of $L$ functions:

$$f(\mathbf{x}) = \sum_{i_{L-1}=1}^{n_{L-1}} \phi_{L-1,i_L,i_{L-1}} \left( \sum_{i_{L-2}=1}^{n_{L-2}} \cdots \left( \sum_{i_2=1}^{n_2} \phi_{2,i_3,i_2} \left( \sum_{i_1=1}^{n_1} \phi_{1,i_2,i_1} \left( \sum_{i_0=1}^{n_0} \phi_{0,i_1,i_0}(x_{i_0}) \right) \right) \right) \cdots \right)$$

where $n_l$ is the number of nodes in the $l^{th}$ layer and $\phi_{l,j,k}$ is the activation function that connects the $k^{th}$ neuron in the $l^{th}$ layer to the $j^{th}$ neuron in the $l+1$ layer. The network architecture is better illustrated in Figure 7 which was adapted from Figure 2.2 in Liu et al. (2024c).

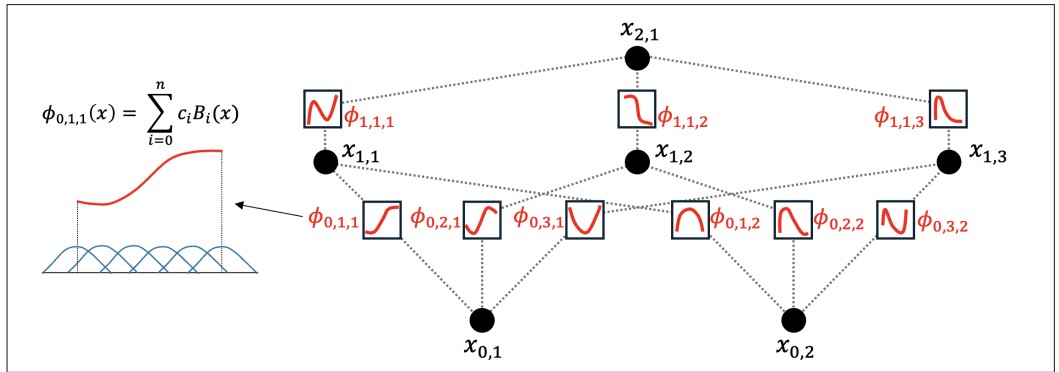

Figure 7: Sample architecture of a Kolmogorov-Arnold Network with three layers of size $[2, 3, 1]$. Activation functions $\phi$ are placed on the edges and are parametrized as a spline. Each output of a node is a sum of its inputs.

Each activation function is comprised of a basis function $b(x)$ and a spline function:

$$\phi(x) = w_b b(x) + w_s \text{spline}(x)$$

where

$$b(x) = \text{silu}(x) = \frac{x}{1 + e^{-x}}$$

$$\text{spline}(x) = \sum_i c_i B_i(x)$$

Initially, $w_s$ is set to 1 and $\text{spline}(x) \approx 0$. The weights of the basis function is initialized according to Xavier initializations.

**KAN Implementation** Although KANs have primarily been applied to science-related tasks such as function approximation and PDE solving, in this work, we adapted the model to address the reconstruction and forecasting tasks posed in the Common Task Framework. The data was processed and loaded similarly to example 14 in the `pykan` repository.

For forecasting tasks, the input-output pairs were constructed in an autoregressive manner, where each input consisted of lagged observations used to predict future values. The input and output dimensions depend on both the number of spatial dimensions in the dataset and the chosen lag (l). By implementing the model in this way, the KAN network approximates the optimal l-step numerical scheme to solve the given dynamical system.

For both the global wavefields and the DAS dataset, if the dataset dimension is $d$ and with a lag of $l$, the input dimension was set to $d_{\text{in}} = d \cdot l$ and the output dimension to $d_{\text{out}} = d$.

For reconstruction tasks, the model was trained in an autoencoding fashion, where each input was mapped directly to itself as the target output: the input and output dimensions were both set to $d_{\text{in}} = d_{\text{out}} = d$.

**Hyperparameters** Based on the hyperparameter settings provided in the `pykan` package and the results reported in the original paper Liu et al. (2024c), the hyperparameters outlined in Tables 16 were selected and tuned for this model. Broadly, the hyperparameters fall into two categories: (1) model architecture and (2) training.

Architecture-related hyperparameters include the number of layers, dimensions of hidden layers, grid resolution, the polynomial degree of the spline basis ($k$), and the lag. Training-related hyperparameters include the number of training steps (epochs), learning rate, overall regularization strength ($\lambda$), and the regularization coefficient for the spline parameters ($\lambda_{coef}$).

| hyperparameter | type | min (or options) | max (or none) |
|---|---|---|---|
| steps | randint | 400 | $10^3$ |
| lag* | randint | 1 | 2 |
| lr | loguniform | $10^{-5}$ | $10^{-1}$ |
| num_layers | randint | 1 | 5 |
| {one-five}_dim** | randint | 1 | 5 |
| grid | randint | 1 | 100 |
| k | randint | 1 | 3 |
| $\lambda$ | loguniform | $10^{-7}$ | $10^{-3}$ |
| $\lambda_{coef}$ | loguniform | $10^{-7}$ | $10^{-3}$ |

Table 16: Hyperparameter search space for the KAN model on the tested datasets. NOTE: The lag parameter is set to zero for reconstruction tasks (pair_id = 2 or 4)*. The dimension of each layer is defined separately. For example the number of nodes in layer two would be defined as *two_dim***.

### A.3.11 NEURAL-ODE

Neural-ODEs are a type of neural network that uses an ODE solver to model the hidden state of a neural network.Chen et al. (2018). This is very similar to ODE-LSTMs, another model evaluated in this work, except it makes use of a vanilla MLP instead of LSTM.

We search over the following hyperparameters: hidden_state_size (dimension of the latent space), seq_length (input sequence length), batch size, and lr (learning rate).

| hyperparameter | type | min (or options) | max (or none) |
|---|---|---|---|
| hidden_state_size | randint | 8 | 1024 |
| seq_length | randint | 5 | 74 |
| batch_size | randint | 5 | 120 |
| lr | log_uniform | $10^{-5}$ | $10^{-2}$ |

Table 17: Hyperparameter search space for Neural-ODE models. We train for 100 epochs.

### A.3.12 LLMTIME

LLMTime Gruver et al. (2023) is a time-series foundation model that uses pre-trained LLMs to perform zero-shot forecasting of time-series data. Their approach is to modify the tokenization of each model so that time-series forecasting is casted as a next-token prediction in text problem. For our evaluation, we used the `llama-7b` as LLMTime's base LLM and used the default temperature of 1.0, an alpha of 0.99, and a beta of 0.3. We also used LLMTime's default Llama tokenizer. LLMTime is only able to forecast univariate time-series, so we auto-regressively forecast each dimension with a context of 200 tokens and a prediction length of 100 tokens at a time. Once each dimension has been forecasted, they are concatenated and evaluated on the test set. For reconstruction tasks, we take the first 10 time-steps of the training data and forecast each dimension until we have a vector containing the same number of timesteps as in the testing dataset and then concatenate and calculate our metrics as before. On all of our datasets LLMTime was unable to generate a complete prediction due to timing out on the 8 hour time limit, thus a -100 score was assigned across all tasks.

### A.3.13 CHRONOS

Chronos Ansari et al. (2024a) is a pre-trained probabilistic time-series foundation model from Amazon. The model is informed by the success of transformers and LLMs, and as such tokenizes time series values using scaling and quantization and trains using the cross-entropy loss function. The model is only capable of doing univariate time-series forecasting. For our evaluation, we use the pre-trained `chronos-t5-base` model and do a one-shot forecast of each dimension of each dataset independently and concatenate them when calculating our metrics. For reconstruction tasks, we take the first 10 time-steps of the training data and forecast each dimension until we have a vector containing the same number of timesteps as in the testing dataset and then concatenate and calculate our metrics as before. Chronos has a much smaller context length than LLMTime due to requiring more VRAM for inference.

### A.3.14 MOIRAI

Moirai_MoE Liu et al. (2024b) is a time-series forecasting foundation model from Salesforce AI Research. The model uses a sparse mixture-of-experts transformer architecture and is able to do one-shot multivariate time-series forecasting on arbitrary time-series datasets. For our evaluation, we used the pre-trained `base` model and predicted 10 time-steps at a time with a context length of 20. For reconstruction tasks, we take the first 10 time-steps of the training data and forecast until we have a matrix containing the same number of timesteps as the testing dataset. Moirai_MoE has a much smaller context length than LLMTime due to requiring more VRAM for inference.

### A.3.15 SUNDIAL

Sundial Liu et al. (2025a) is a family of native, flexible and scalable time-series foundation models from Tsinghua University, tailored specifically for time series analysis. It is pre-trained on TimeBench (about one trillion time points), adopting a flow-matching approach rather than fixed parametric densities. Sundial directly models the distribution of next-patch values in continuous time-series without discrete tokenization; it is built on a decoder-only Transformer architecture. For our evaluation, we used the pre-trained `sundial-base-128m` model; the model can handle multivariate time-series forecasting directly. For datasets tested, due to RAM limitations, we have split the "spatial" dimension into batches, forecasting each batch independently and concatenating the results. For reconstruction tasks, we take some of the first time-steps of the training data (around 10%) and forecast until we have a matrix containing the same number of timesteps as the testing dataset.

### A.3.16 PANDA

Panda Lai et al. (2025) is a foundation model for nonlinear dynamical systems based on Patched Attention for Nonlinear Dynamics. Panda is motivated by dynamical systems theory and adopts an encoder-only architecture with a fixed prediction horizon. It is pre-trained purely on a synthetic dataset of $2 \times 10^4$ chaotic dynamical systems, discovered using a structured algorithm for dynamic systems discovery introduced in the same work. For our evaluation, we used the pretrained model weights provided on the official code repository associated with Lai et al. (2025). The main free parameter in the forecasts with Panda is the context length. Due to RAM limitations for the datasets tested, we have to limit the context to 512 observations.

### A.3.17 TABPFN-TS

TabPFN for Time Series (TabPFN-TS) Hoo et al. (2025) is based on the tabular foundation model TabPFNv2 Hollmann et al. (2025), adapted to the task of time series forecasting. We use the pretrained model weights, leaving the only remaining parameter as the amount of data for each specific system that the model is exposed to before performing zero-shot forecasting. For the global wavefields and DAS datasets, we restrict to at most 500 time steps to be used for context. This restriction was introduced as a result of limited available memory, and is similar to the restriction placed on Panda.

