# OpenReview forum: "The Seismic Wavefield Common Task Framework"
_ICLR.cc/2026/Conference — ICLR 2026 Poster_

### Official Review · Reviewer_heXL · 2025-10-29

**Soundness:** 2
**Presentation:** 1
**Contribution:** 3
**Rating:** 4
**Confidence:** 4

**Summary:**

This paper introduces a Common Task Framework (CTF) for benchmarking machine learning methods on seismic wavefield modeling tasks. The framework aims to provide standardized evaluation and fair comparison in seismology with forecasting, reconstruction, and generalization problems. It includes three datasets (global wavefields, crustal simulations, and fiber-based DAS recordings), defines twelve evaluation metrics, and compares 14 different highly-cited ML models.

**Strengths:**

1. The paper introduces a benchmark suite for seismology, trying to address reproducibility and comparability challenges in scientific problems.

2. It offers a diverse range of datasets, covering different scales and difficulties.

3. It provides a comprehensive metric design with 12 evaluation metrics, which reflect multiple facets of the modeling: forecasting accuracy, robustness to noise, data limitation, and parametric generalization.

4. The paper provides a benchmark including 14 different methods.

**Weaknesses:**

1. The paper uses inconsistent names: “ctf4seismology,” “Seismic Wavefield CTF,” and “seismo dataset.” This inconsistency is confusing.

2. The details of the datasets are unclear. A summary table should formally describe each dataset's size, format, train/validation/test splits, etc.

3. Important details are missing for each task, such as noise levels, limited data size (M), and number of forecasted time steps (m).

4. The paper would benefit from data visualizations and formal descriptions to help readers better understand the data and tasks. In particular, a more detailed explanation of the underlying physics is needed to highlight what makes seismic wavefield data different from general time-series data. Section 2.1 (called “Seismic Wavefields as Spatio-Temporal Systems”) does not clearly show what the system is and how spatial and temporal structures look like. To encourage broader community participation in the proposed CTF, the description should be more helpful to those non-seismologists in understanding the physics and challenges of the data.

5. No training hyperparameters and model settings are provided for the evaluated methods.

6. Table 1, Figures 3 and Figure 4 show almost the same information in different formats, consuming space without adding insight.

Overall, I appreciate the work to propose a CTF for a scientific problem. However, the paper currently reads more like an introduction or announcement of the upcoming Kaggle competition rather than a formal scientific paper. Many details are missing, making it hard to fully assess the paper.

**Questions:**

Please refer to the Weakness

---

> ### Author Response · Authors · 2025-11-22
> **Response to Reviewer heXL**
>
> ## General Response
>
> Please see our general response to all reviewers as a top-level comment in our Open Review page.
>
> ## Response to Weaknesses:
>
> 1)
> Thank you for bringing this up. We agree that it is important that the paper uses consistent naming throughout to reduce confusion. The full name of the framework is “The Seismic Wavefield Common Task Framework” which we abbreviated “ctf4seismology”. However, we understand this can be confusing, so we have consolidated all references to the CTF to just be “The Seismic Wavefield Common Task Framework”. Furthermore, “Common Task Framework” is abbreviated “CTF”, which we left in the manuscript. Lastly, we have removed reference to “seismo dataset”. The name “seismo” is the name of the global wavefields dataset in our Github repository, but we agree that making this clarification in the paper is not necessary. Instead, we leave the reference to the Github repository and leave it up to the reader to visit it and read the documentation to understand where the different datasets are located and how to use our framework and API with their own models.
>
> 2)
> Thank you for this important comment. We want to ensure that the datasets make sense and can be understood at a glance from the paper. As a result, we have added summary tables for dataset sizes, formats, and how the train/validation/test splits are generated. These changes are reflected in the updated manuscript.
>
> 3)
> Thank you for bringing this to our attention. In response to your previous question, we have added the dataset sizes, including values for (M) and (m), in our updated manuscript. As for noise levels, we want to reiterate that the goal of our CTF is to provide fair and reliable assessments of machine learning models on seismic wavefield datasets. As such, for the same reason we have a hold-out test set, we chose to omit the amount of noise added to the training data. If noise levels were provided, machine learning models would not be needed and classical numerical approaches could be used instead. We hope you understand the importance of omitting the noise levels in our CTF. We have used our expert judgement to determine what constitutes low- and high-noise during dataset creation.
>
> 4)
> Thank you for raising this point. We have updated the manuscript to make more clear the visualizations in Figure 1 and have updated the body of the text to further describe the datasets we highlight as part of our CTF. We go into more detail on the physics and the spatial and temporal structures. We hope this makes more clear to non-seismologists what the datasets look like and their unique characteristics.
>
> 5)
> Thank you for this important point. We have added extensive information about training hyperparameters for the evaluated methods to the appendix of our updated manuscript. We apologize for omitting this during the initial submission.
>
> 6)
> Thank you for bringing this to our attention. We have consolidated Figures 3 and 4 into Table 1, which provides the same amount of information but reduces consumed space. We have filled in the additional space with dataset details and more thorough descriptions of the datasets.
>
> ## Response to Overall Purpose of the Work
>
> Thank you for this comment. We hope that by addressing your questions and weaknesses above that we have successfully convinced you of the importance of our work and how it fits into the ML and Seismology communities and have filled in the necessary details to make our paper worthy of publication. Thank you for your continued time and effort in reviewing our manuscript.

---

> > ### Author Response · Authors · 2025-11-22
> > **Response to Reviewer heXL (continued)**
> >
> > We hope that our updated manuscript in response to your questions and comments addresses your concerns, and thereby encourages you to consider raising your score for acceptance. Thank you again for your time, effort, and consideration of our manuscript.

---

> ### Comment · Reviewer_heXL · 2025-11-26
>
> Thanks to the authors’ response, my questions were answered. I will increase my score to 6.

---

### Official Review · Reviewer_97i9 · 2025-10-31

**Soundness:** 3
**Presentation:** 3
**Contribution:** 3
**Rating:** 8
**Confidence:** 3

**Summary:**

This paper introduces a Common Task Framework (CTF) for machine learning in seismology, specifically targeting the modeling and reconstruction of seismic wavefields. The authors present three curated datasets (global, crustal, and local scales) and 12 task-specific metrics, aiming to standardize evaluation and comparison of ML methods in this domain. The framework is inspired by successful CTFs in other fields (e.g., ImageNet) and is demonstrated through benchmarking a variety of ML models on the global wavefields dataset.

**Strengths:**

1. Meaningful Contribution:
Seismology is a domain where ML adoption is accelerating, but progress is hampered by inconsistent evaluation, weak baselines, and reporting bias.
2. Well-Designed Evaluation Protocol:
The authors propose a multi-metric scoring system that captures different aspects of model performance (forecasting, reconstruction, noise robustness, limited data, parametric generalization).
3. Benchmarking and Transparency:
The paper benchmarks a diverse set of ML models (RNNs, neural operators, DMD, etc.) and provides detailed results, including cases where complex models fail to outperform simple baselines. This transparency is valuable for the community and highlights the challenges ahead.
4. Open Platform and Reproducibility:
The datasets, code, and evaluation scripts are made publicly available, with plans for a Kaggle competition. This openness will foster community engagement and accelerate progress.

**Weaknesses:**

Performance and Impact:
The results show that most ML models struggle to outperform trivial baselines on the provided tasks. While this highlights the difficulty and the need for better methods, it also suggests that the immediate impact of the platform may be limited until more sophisticated models or richer datasets are available. It will be helpful to classify the dataset based on the difficulty level, so that the ML area can design solutions from easy to difficult task gradually.

**Questions:**

N/A

---

> ### Author Response · Authors · 2025-11-22
> **Response to Reviewer 97i9**
>
> ## General Response
>
> Please see our general response to all reviewers as a top-level comment in our Open Review page.
>
> ## Response to Weaknesses:
>
> 1)
> Thank you for your generous review of our manuscript. We appreciate you bringing up this weakness. We have consulted with expert seismologists and real ML practitioners to update the manuscript to speak more on the complexities of each dataset. We hope that with this change and the other changes from the other reviewers, our manuscript will be a valuable contribution to the seismology community that are interested in data-driven modeling of both simulated and real-world data.

---

### Official Review · Reviewer_6Sog · 2025-11-01

**Soundness:** 1
**Presentation:** 2
**Contribution:** 1
**Rating:** 2
**Confidence:** 3

**Summary:**

This paper proposed a Common Task Framework (CTF) to evaluate machine learning algorithms for seismic wavefields. Three distinct datasets across various scales are included. The authors also established a series of tasks and proposed their corresponding evaluation metrics. They further provided the benchmarking results of 16 methods on the global wavefield dataset.

**Strengths:**

The authors provided helpful descriptions of the models and the seismic wavefields in the datasets.

**Weaknesses:**

While the idea of establishing a CTF is interesting, I feel that the current work has limited novelty and the experimental evaluation appears incomplete. My details comments are as follows.
1. Although three datasets are included in the proposed CTF, the authors did not clearly explain how these datasets relate to one another or why they should be integrated into a single framework.
2. The authors didn’t highlight the challenges involved in generating these datasets. For instance, the global wavefield dataset is generated using a public Earth model, and the crustal dataset is an extension of the existing work. Besides, the volume of each dataset is also relatively small.
3. In the experiment section, the authors presented benchmarking results only for the global wavefield dataset, and the results on the other two datasets are missing.
4. The related works section is missing.

**Questions:**

1. For the DAS dataset, could the authors provide a description of the models used for the generation of wavefields?
2. In the experiment section, could the authors provide some visualizations of the predictions of various methods to give readers more intuition about tasks and model performance?
3. [Minor] Figure 1 is organzied in a different order from the introduction of three datasets in Section 2.1.

---

> ### Author Response · Authors · 2025-11-22
> **Response to Reviewer 6Sog**
>
> Response to Reviewer BaxP
>
> ## General Response
>
> Please see our general response to all reviewers as a top-level comment in our Open Review page.
>
> ## Response to Weaknesses:
>
> 1)
> We appreciate the opportunity to clarify why a single framework is necessary and why we selected these three datasets. A single unified CTF for seismology allows for a set of standardized evaluation protocols and metrics that give practitioners insight into which models excel at specific tasks and wavefields. Having separate, isolated frameworks would further fragment the current ML literature in seismology, hindering progress towards effective seismic ML models. Our Github repository and Kaggle competition invites contributors to evaluate new models on our datasets as well as provide new datasets to the community, all within the same framework.
>
> The selected datasets represent three interesting datasets that our professional seismologist collaborators would like to see evaluated on ML models. These datasets represent a wide range of seismic wavefield scales (global, crustal, and local) that real practitioners are interested in.  Importantly, we need a beginning point to start the CTF and we have judiciously selected three data sets that are representative of the challenges the community faces in characterizing and learning seismology.
>
> 2)
> Thank you for this important observation. We have updated the manuscript to highlight the challenges in dataset collection and generation. Furthermore, we respect the point that the volume of the datasets are relatively small. This was a design choice to democratize participation in the CTF. Too often is machine learning hindered by data storage and computation, making the only viable competitors large-scale companies like Google, Meta, and OpenAI. In contrast, our code and datasets can be run from consumer laptops. We hope that our CTF invites participation from all communities interested in driving forward the technical innovations required to improve ML models on seismic wavefield data. As our CTF is expandable, we will also look into higher-scale datasets for future Kaggle competitions.
>
> 3)
> Thank you for raising this point. We partially address your comment in the general response above. As we mentioned, the purpose of our CTF is not to be exhaustive, but a scalable foundation on top of which seismology will be able to evaluate current and future ML models on a wide range of seismic datasets. Nonetheless, we agree that the introduction of our framework should include more results to make a stronger first-impression to generate community engagement. As a result, we have performed extensive evaluation on the real-world Distributed Acoustic Sensing dataset. This dataset complements the simulated seismic wavefields dataset in our original manuscript by coming from different geometry (linear vs spherical), coming from real-world sensor measurements, different scales (local vs global) and utilizing a modern technology that is already being used with machine learning models [1] [2] [3]. We have also added two more foundation models to our results: Sundial and LLMTime. Please refer to the manuscript for a detailed description on the DAS dataset as well as for model performances on this dataset.
>
> We also have the 3D crustal wavefield dataset ready for the Kaggle competition and are eager to see how the community responds to this challenging dataset.
>
> [1] Ni, Yiyu, et al. "Wavefield reconstruction of distributed acoustic sensing: Lossy compression, wavefield separation, and edge computing." Journal of Geophysical Research: Machine Learning and Computation 1.3 (2024): e2024JH000247.
>
> [2] Stork, Anna L., et al. "Application of machine learning to microseismic event detection in distributed acoustic sensing data." Geophysics 85.5 (2020): KS149-KS160.
>
> [3] Hernandez, Pablo D., Jaime A. Ramirez, and Marcelo A. Soto. "Deep-learning-based earthquake detection for fiber-optic distributed acoustic sensing." Journal of Lightwave Technology 40.8 (2021): 2639-2650.
>
> 4)
> We appreciate this comment. While it’s true we did not have a related works section labeled, we talk extensively about related works in section 1. We have adjusted the manuscript to make the related works section more explicit.

---

> > ### Author Response · Authors · 2025-11-22
> > **Response to Reviewer 6Sog (continued)**
> >
> > ## Response to Questions:
> >
> > 1)
> > Thank you for this question. We have added a more thorough description of the DAS dataset in the updated manuscript.
> >
> > 2)
> > Thank you for this question. We agree that visualizations are a useful way to demonstrate the performance of the models. There are many ways to visualize our results and we’re not sure exactly what you would find most useful as a reader. As an example, here is a link to some sample forecasts for the best models on the global wavefields dataset: https://imgur.com/a/qYMInbu (please click the images to see the movies). We plotted the results of the high-noise long-term forecasting task (E6) from the Reservoir model, which scored the highest on any long-term forecasting task. We also plotted the results of the short-term no-noise forecasting task (E1) and short-term low-noise forecasting task (E9) from the Chronos foundation model, which scored the highest on those short-term forecasting tasks. Notice that even the best forecasting models on the global seismic waves dataset do not provide any visual insight, predicting zeros or the last training timestep. Due to a lack of any of the models performing well in addition to the page limit, we omitted these plots in our original submission. We have updated the revised manuscript to mention this in the main content of the paper.
> >
> > Regardless, we want to further drive transparency and openness in our CTF and would like to incorporate your suggestion. We have several ideas of what we could do, but would like to have your input on what’s most useful as a reader. 1) We can provide the plots from the above link (https://imgur.com/a/qYMInbu) in the appendix for the best performing models. 2) We can provide plots like https://imgur.com/a/zbYcP8m where we show the forecasts for single sensor measurements on the global wavefields dataset. 3) We can provide plots for either of the two previous formats in the Kaggle competition, so that during the competition the community as a whole can see how each model performs and what it might be doing well on. 4) We can add a generated plots directory to our public Github which would contain a database of all model predictions for each task. Please let us know what is most informative and useful!
> >
> > 3)
> > Thank you for this comment. The textual description of the datasets in Section 2.1 has been updated to match the order of the figures in Figure 1.

---

> > > ### Author Response · Authors · 2025-11-22
> > > **Response to Reviewer 6Sog (continued)**
> > >
> > > We hope that our updated manuscript in response to your questions and comments addresses your concerns, and thereby encourages you to consider raising your score for acceptance. Thank you again for your time, effort, and consideration of our manuscript.

---

### Official Review · Reviewer_BaxP · 2025-11-01

**Soundness:** 3
**Presentation:** 2
**Contribution:** 3
**Rating:** 6
**Confidence:** 4

**Summary:**

This work presents a new common task framework for seismic wavefield estimation tasks. The framework seems to be hosted on github, and is proposed to be hosted on Kaggle platform in future. The framework currently seems to incorporate three global scale datasets from seismology domains. The tasks are defined with multiple perspectives with 12 subtasks and scoring metrics. Surprisingly, modern models tend to perform worse than zero-knowledge baseline model of predicting only zeros of all tasks.

**Strengths:**

This manuscript is presenting a new aggregated datasets and unifying evaluation system for tasks related to seismic wavefields. Seismology contains diverse tasks and domains, yet the proposed task has very large scale (i.e. planetwise), and immediate extensibility from Earth to Mars. The three datasets proposed seem to be well accepted and analyzed by domain experts. As an ML expert with seismological application research experience, I see the value underlying this work as a domain-expert-curated dataset-evaluation combination which takes care of two key issues: 1) is the data trustworthy, and 2) is the metric meaningful. These two questions had been and will be of greatest concern for a serious AI/ML driven interdisciplinary research.

**Weaknesses:**

Seismology contains lots of distinctive tasks from various aspects of solid earth (be it from the Earth, Moon, or Mars). Planet-wide seismic wavefield tasks, which are the primary focus of this manuscript, represent a portion of the wide field. The claimed name 'ctf4seismology' may be what the authors want to reach, but as of current form, the name is too assuming and overstating what the current dataset-evaluation-task proposal contains. I think one more spoonful of factual representation would be a good addition.

Also, perhaps due to the page limit, the materials presented in the 9 page limit did not provide sufficient information for me to gauge what would be a reasonable method and how to use the framework if I have a suitable method. This is the perspective of ML practitioner, who'd like to get an idea of 'how I can use it'. If there is a public repository, it would be much better to have a link to it explicitly shared in the manuscript.

**Questions:**

Are the twelve tasks equally important for seismic wavefield research? The current proposal seems to suggest a simple averaging of all subscores --- but I suspect that depending on the research direction some tasks will have greater relevance than others. Is this something left for seismologists? As it is presented in the current form, it seems like a call for ML practitioners to play with this new set of tasks.

Is there a publicly available repository to share the baseline codes and/or solicit collaboration, as alluded by the future directions given in the text? Would the proposed CTF extend well to incorporate wide range of distinct data-driven tasks in seismology (not only global but also local and/or heterogeneous)? This question may be the most important one if the authors plan to claim the audacious name of 'ctf4seismology'.

---

> ### Author Response · Authors · 2025-11-22
> **Response to Reviewer BaxP**
>
> ## General Response
>
> Please see our general response to all reviewers as a top-level comment in our Open Review page.
>
> ## Response to Weaknesses:
>
> 1)
> Thank you for this important point. We agree that there are many different aspects of solid earth dynamics/physics and that a Common Task Framework for Seismology as a whole should encapsulate a wide range of those aspects. We partially address your comment in the general response above. That being said, we agree that expanding the benchmark is important for broader scientific relevance and plan on adding more datasets relevant to seismology over time while also recognizing that seismology is a large field and a subset of representative datasets must be selected for the CTF as a starting point. To partially address your concerns, we added extensive evaluation on the real-world Distributed Acoustic Sensing (DAS) dataset. This dataset complements the simulated seismic wavefields dataset in our original manuscript by coming from different geometry (linear vs spherical), coming from real-world sensor measurements, different scales (local vs global), and utilizing a modern technology that is already being used with machine learning models [1] [2] [3]. Please refer to the updated manuscript for a detailed description on the DAS dataset as well as for model performances on this dataset.
>
> [1] Ni, Yiyu, et al. "Wavefield reconstruction of distributed acoustic sensing: Lossy compression, wavefield separation, and edge computing." Journal of Geophysical Research: Machine Learning and Computation 1.3 (2024): e2024JH000247.
>
> [2] Stork, Anna L., et al. "Application of machine learning to microseismic event detection in distributed acoustic sensing data." Geophysics 85.5 (2020): KS149-KS160.
>
> [3] Hernandez, Pablo D., Jaime A. Ramirez, and Marcelo A. Soto. "Deep-learning-based earthquake detection for fiber-optic distributed acoustic sensing." Journal of Lightwave Technology 40.8 (2021): 2639-2650.
>
> 2)
> You raise a valid concern. We agree that both ML practitioners and Seismologists should be able to look at our paper and gauge which method is reasonable to use for a specific problem as well as how to use the framework to evaluate their own method. We have a well-documented and public GitHub repository, and we made the link to it more clear in our revised manuscript. The repository contains detailed instructions on how to use the framework and how to extend it to use a new dataset and/or new model. Furthermore, we have added some text to the section “3.1 Observations” providing guidance to practitioners on which model to use for their task. As mentioned in response to your question about the weight of each task, it’s ultimately up to the practitioner to decide which task metric to focus on. We provide a holistic approach by scoring a variety of tasks, but there isn’t a one-size-fits-all approach to distilling those tasks into a single metric that can be used by everyone. We make this point clear in our revised manuscript.
>
> ## Response to Questions:
>
> 1)
> Excellent observation. The CTF we present is meant to provide a holistic view for each evaluated model in Seismology, which motivates the use of twelve tasks. Each task represents a particular use-case that a model might be used for, from de-noising to long-term forecasting, etc.. Thus, your point is exactly right: for different research directions (even within specific use-cases in seismology), some tasks are more relevant than others. There is no way to know this apriori and so we compromise and use an average of metrics when a single score is requested. By providing the results for each model across all twelve metrics, we leave it up to the practitioner to identify which model they would like to use based on the best-performing models using the specific metrics of interest. This recommendation is reflected in the updated manuscript.
>
> 2)
> Thank you for this question. Yes, there is a publicly available repository with all code needed to reproduce our results and to solicit collaboration. We make the link to the Github repository more clear in our revised manuscript. Please note, that we are currently using an anonymized link so the domain is not github.com, but rather anonymous.4open.science. Our code is easily extendable to allow for quick adoption of new models and datasets, providing a solid foundation for our scalable framework.

---

### Author Response · Authors · 2025-11-22

## General Response to the Reviewers

We thank the reviewers for their thorough assessment of our manuscript and their insightful feedback. We are grateful for the opportunity to clarify the scope and ambitions of our work. We believe our responses below and the revisions in the manuscript will address all the concerns raised.

The primary objective of the Seismic Wavefield CTF is to establish a unified evaluation framework for machine learning (ML) in seismology.  Currently, AI methods are being broadly deployed in seismology for a variety of data sets without any clear way to assess how they compare with baselines or competing architectures. This initiative provides a standardized infrastructure for formatting seismic datasets and evaluating models against a diverse suite of metrics. Our initial release, featuring a curated selection of highly-cited models evaluated on key seismic datasets, is designed not as an exhaustive survey, but as a scalable foundation for the community. The field of seismology is vast and cannot be encapsulated by any single effort; our framework is intended to expand, prioritizing datasets and models that reflect the most pressing and trending research directions in the community.  Importantly, the CTF provides a critical educational element for students, postdocs and faculty to understand the effectiveness of diverse AI methods for characterizing and advancing the field of seismology.

To catalyze this growth, we will leverage community-driven platforms. On one hand, we have developed robust and well-documented code that is open-source, allowing anyone to implement their own models and datasets to encourage quick adoption. On the other hand, we are preparing a Kaggle competition to generate widespread engagement which will be pre-populated with the results presented in this paper. Over time, we anticipate that grassroots efforts from the seismology community will expand the framework, implementing new models and integrating new datasets. This addresses a critical need: it provides a data-driven method for seismologists to select models based on empirical performance for their specific application, moving beyond reliance on citation counts or ad-hoc selection.
In direct response to the reviewers' emphasis on evaluating more datasets, we have already expanded our framework by scoring the real-world Distributed Acoustic Sensing (DAS) dataset. This addition demonstrates our commitment to including complex, real-world seismic data that challenges existing ML models. The integration of this dataset underscores the practical applicability of the CTF and its readiness to support community-driven ML evaluation efforts. Furthermore, we have scored two additional foundation models, Sundial and LLMTime, on the global wavefields and DAS datasets.

Below, we address your specific comments and provide a current version of the scores on the DAS dataset. We are currently hyperparameter tuning the rest of the models and will let all the reviewers know of when all the models are evaluated on the DAS dataset in the updated manuscript. Please refer to the updated manuscript for our additions and changes based on your feedback. Thank you for your time, effort, and consideration of our manuscript.

---

> ### Author Response · Authors · 2025-11-25
> **DAS Evaluation Completed**
>
> Dear reviewers,
>
> Thank you for your patience. We have finished hyperparameter tuning and evaluation on the DAS dataset. For our rebuttal, we have added the LLMTime, Pandas, and Sundial foundation models, putting us up to 17 models (+2 baselines) evaluated on 12 tasks across 2 datasets.
>
> Thank you.

---

### Author Response · Authors · 2025-12-02
**Author Final Remarks for Discussion Period**

## Reviewers:

Thank you for your invaluable input, time, and consideration of our manuscript. We wish that we would have been able to continue the discussion the intended way. Nevertheless, we are grateful for the discussions we were able to have and the improvements we have made with your help. Our work is undoubtedly strengthened by your input.

## Area Chairs:

We hope you find our proposed Seismic Wavefield Common Task Framework (CTF) as an important contribution to the intersection of seismology and machine learning. Our open-source initiative provides a scalable, community-expandable foundation starting with curated models and expert-selected datasets. Through the end of the discussion period we have hyper-parameter tuned and evaluated 17 models on 2 datasets across 12 different metrics. The datasets span different scales (global vs local) and geometries (spherical vs linear) of interest to seismologists. To generate community engagement, we will be launching a Kaggle competition that utilizes our robust, open-source, unit-tested, and documented code. In the long run, we see our Seismic Wavefield CTF as a critical component in advancing machine learning in seismology, much like what has been done in reinforcement learning, computer vision, and LLMs.

During the rebuttal period, we have responded comprehensively to every comment and question. Two of the lower scores were influenced by information that was present but may have been overlooked, as well as by misunderstandings of our methods and results. We have clarified these points in detail, along with providing an entire new set of results and significantly improving our manuscript. We believe the revised explanations substantially mitigate the original concerns. We also highlight that Reviewer heXL has already increased their score from 4 to 6 after considering our responses. We hope these clarifications assist the area chairs in creating a well-informed final decision.

---

### Meta-Review · Area_Chair_tPnm · 2026-01-05

**Summary:**

This paper introduces the Seismic Wavefield Common Task Framework, aiming to standardize evaluation of ML methods for seismic wavefield modeling with curated datasets, tasks, and metrics, plus an open-source implementation and planned community engagement (e.g., Kaggle).
Reviewers agreed the idea is valuable for the seismology/ML intersection, but early concerns focused on missing/unclear dataset and task details, insufficient breadth of evaluation beyond the main dataset, and whether the submission reads more like a platform announcement than a complete paper. The rebuttal added substantial missing content (notably additional evaluation and clearer documentation), shifting the balance toward acceptability as a benchmark/framework paper.

**Reviewer Concerns:**

Addressed by rebuttal:
- Completeness of evaluation: Authors expanded results (notably adding DAS evaluation and more models), reducing the “only one dataset evaluated” concern.
- Clarity/details: Added clearer dataset/task descriptions (tables, splits, parameters), training hyperparameters (appendix), and improved naming consistency; one reviewer explicitly raised their score after these changes.
- Related work / positioning: Made related-work discussion more explicit and clarified the scope and intent (foundation to be expanded by the community).

Still outstanding:
- Perceived novelty/impact: One reviewer remains skeptical that the work is sufficiently “research-y” versus infrastructure/announcement, and the fact that many models fail against simple baselines may limit near-term impact (though it is also an honest and useful finding).
- Scope/title expectations: Even with clarifications, the long-term “CTF for seismology” framing can still read broader than what the current datasets cover.

**Reviewer Scores:**

My estimate if reviewers had fully participated in discussion:
- BaxP (6): likely stays 6.
- 6Sog (2): likely increases to 3–4 (many completeness/related-work/detail issues were addressed; novelty concern may remain).
- 97i9 (8): likely stays 8.
- heXL (raised to 6): already updated to 6.

---

### Decision · Program_Chairs · 2026-01-26

Accept (Poster)